# Childhood body size directly increases type 1 diabetes risk based on a lifecourse Mendelian randomization approach

Tom G. Richardson [1,2 ✉], Daniel J. M. Crouch[3], Grace M. Power [1], Fernanda Morales-Berstein [1], Emma Hazelwood [1], Si Fang[1], Yoonsu Cho [1], Jamie R. J. Inshaw[3], Catherine C. Robertson [4], Carlo Sidore[5], Francesco Cucca [5], Steven S. Rich [4], John A. Todd [3] & George Davey Smith [1]

The rising prevalence of childhood obesity has been postulated as an explanation for the increasing rate of individuals diagnosed with type 1 diabetes (T1D). In this study, we use Mendelian randomization (MR) to provide evidence that childhood body size has an effect on T1D risk (OR = 2.05 per change in body size category, 95% CI = 1.20 to 3.50, P = 0.008), which remains after accounting for body size at birth and during adulthood using multi-variable MR (OR = 2.32, 95% CI = 1.21 to 4.42, P = 0.013). We validate this direct effect of childhood body size using data from a large-scale T1D meta-analysis based on n = 15,573 cases and n = 158,408 controls (OR = 1.94, 95% CI = 1.21 to 3.12, P = 0.006). We also provide evidence that childhood body size influences risk of asthma, eczema and hypo-thyroidism, although multivariable MR suggested that these effects are mediated by body size in later life. Our findings support a causal role for higher childhood body size on risk of being diagnosed with T1D, whereas its influence on the other immune-associated diseases is likely explained by a long-term effect of remaining overweight for many years over the lifecourse.

[1] MRC Integrative Epidemiology Unit (IEU), Population Health Sciences, Bristol Medical School, University of Bristol, Oakfield House, Oakfield Grove, Bristol, United Kingdom. [2] Novo Nordisk Research Centre Oxford, Old Road Campus, Oxford, United Kingdom. [3] JDRF/Wellcome Diabetes and Inflammation Laboratory, Wellcome Centre for Human Genetics, Nuffield Department of Medicine, NIHR Biomedical Research Centre, University of Oxford, Oxford, United Kingdom. [4] Center for Public Health Genomics, University of Virginia, Charlottesville, VA, United States. [5] Institute for Research in Genetics and Biomedicine (IRGB), Sardinia, Italy. ✉email: Tom.G.Richardson@bristol.ac.uk

The incidence of type 1 diabetes (T1D) has doubled in the last 20 years. Possible explanations for this increasing T1D burden include secular changes to gut microbiota linked to the hygiene hypothesis in which increased sanitation[1], urban living and other factors contribute to increases in not only T1D but in a number of other immune system-related diseases, such as multiple sclerosis and asthma[2]. Additional explanations for this increasing burden include the association of virus infection with T1D[3] and decreasing levels of vitamin D in the population[4]. One hypothesis, supported by some observational studies[5,6], is that the rising prevalence of childhood obesity in an increasingly obesogenic environment[7–9], including poor diets with high fat, salt and carbohydrate, may contribute towards early life β-cell fragility and increased susceptibility to T1D[10]. Developing insight into the contribution of childhood body size to T1D risk is extremely challenging, however, particularly in terms of separating its effect from early life confounding factors such as birthweight[11].

In contrast to T1D, there is irrefutable evidence that children who are overweight are more likely to develop type 2 diabetes (T2D) and that weight loss can lead to its sustained remission[12]. We recently used human genetic data to infer that this relationship is likely to be causal rather than due to confounding factors, using sets of genetic variants which robustly associate with childhood and adulthood body size[13]. This was achieved using Mendelian randomisation (MR), which can be implemented through instrumental variable analysis, exploiting the quasi-random assortment of genetic alleles at birth to infer causality between lifestyle exposures and disease outcomes[14–16].

We showed previously that childhood body size increases T2D risk when analysed in a univariable setting (Odds Ratio (OR) = 2.32, 95% confidence interval (CI) = 1.76 to 3.05, $P = 3.83 \times 10^{-9}$)[13]. This approach to estimating the total effect of childhood body size on the risk of disease is presented in Fig. 1A. However, by simultaneously estimating the genetically predicted effects of childhood body size and adulthood body size as separate exposures to T2D risk using a multivariable model, the childhood estimates attenuated to include the null (OR = 1.16, 95% CI = 0.74 to 1.82, $P = 0.52$). As such, there is considerably weaker evidence that childhood body size has a 'direct effect' on T2D risk, as compared to it having an 'indirect effect' mediated via adult body size. Diagrams illustrating how multivariable MR can be applied to estimate direct and indirect effects can be found in Fig. 1B, C respectively. These results, therefore, suggest that the univariable estimates for childhood body size can be explained by long term, persistent effects of adiposity due to individuals typically remaining overweight into adulthood.

Although childhood body size has been previously implicated in T1D risk using MR[17,18], these findings were based on effect estimates derived using a small number of instruments (n = 23). Furthermore, multivariable analyses in this study did not investigate the direct and indirect effects of potential confounding factors which may be pleiotropically influenced by genetic instruments for the exposure of interest. This is particularly important, as exemplified by the case of high density lipoprotein cholesterol onto coronary heart disease risk, which appears to have a protective effect in a univariable setting (OR = 0.80, 95% CI = 0.75 to 0.86, $P = 1.66 \times 10^{-10}$), but not when assessing its direct effect after taking into account atherogenic lipoprotein lipid traits (OR = 0.91, 95% CI = 0.74 to 1.12, $P = 0.36$)[19]. Lastly, it has not yet been investigated whether the effect of childhood body size on T1D risk represents a more generalisable effect on the immune system which may additionally impact other types of immune-associated or autoinflammatory diseases. If there is a T1D-specific effect, this would suggest early life β-cell fragility stemming from diet-induced metabolic stress is likely to be a causal pathway through which childhood body size leads to increased T1D risk.

Consequently, in the present study we had four aims:

1. Investigate evidence of a direct effect of childhood body size on T1D risk by conducting univariable and multivariable

**Univariable Mendelian randomization**

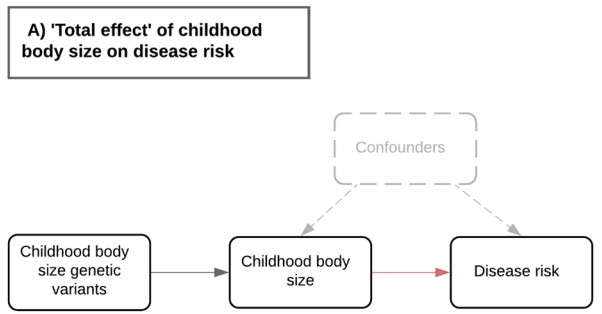

**Multivariable Mendelian randomization**

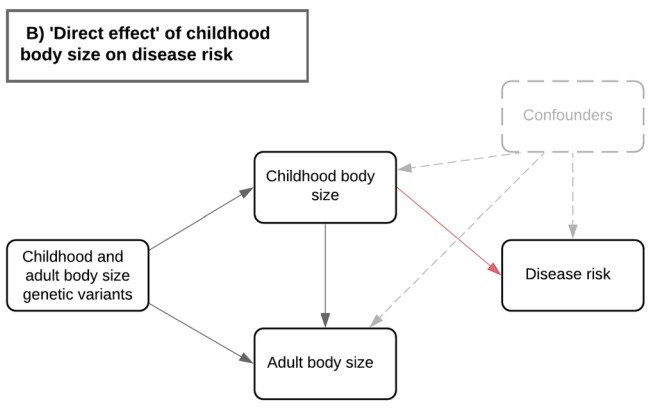

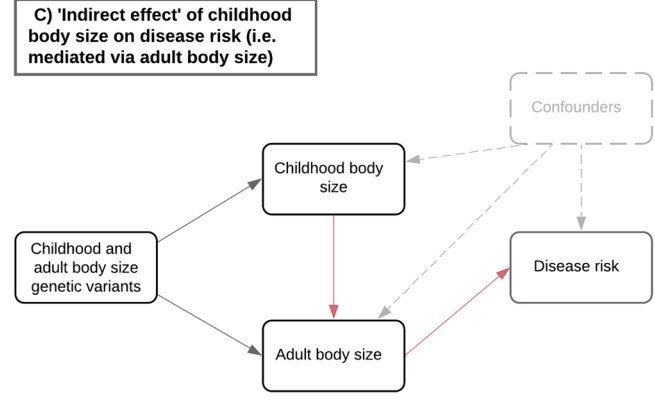

**Fig. 1 Directed acyclic graphs depicting the effects of childhood body size on disease risk.** Schematic representation of the analysis undertaken in this study using Mendelian randomisation (MR). **A** Using univariable MR to estimate the total effect of genetically predicted childhood body size on type 1 diabetes (T1D) risk without accounting for adulthood body size. **B** Applying multivariable MR to estimate the direct effect of genetically predicted childhood body size on T1D risk whilst accounting for the effect of adult body size and **C** using the same approach to estimate the indirect effect of childhood body size of T1D (via adult body size). The highlighted red lines indicate the causal pathway being evaluated in MR to estimate the **A** total, **B** direct, and **C** indirect effects of childhood body size on T1D risk.

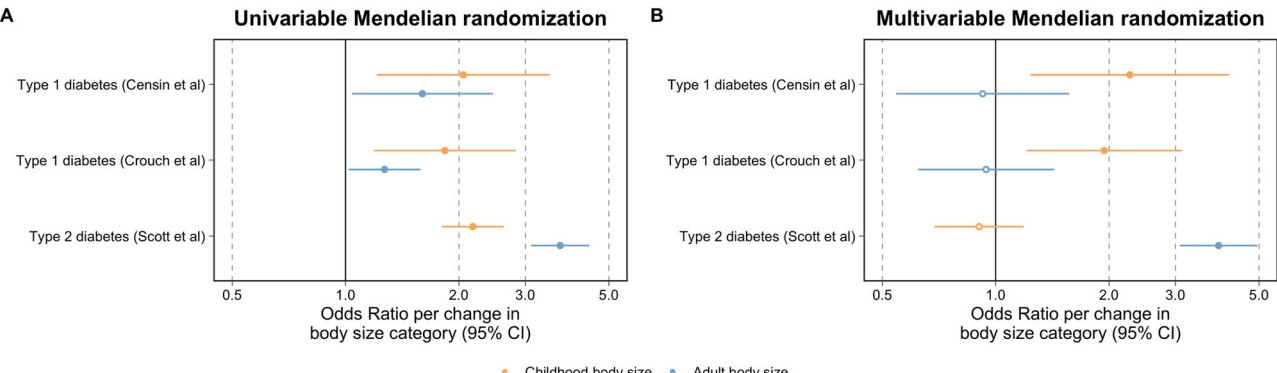

**Fig. 2 Forest plots illustrating the total and direct effects of childhood body size on type 1 and type 2 diabetes risk. A** The univariable Mendelian randomisation (MR) estimates between childhood (yellow) and adult (blue) body size ($n = 453,169$) on risk of type 1 (using estimates from both discovery ($n = 14,741$) and replication analysis ($n = 173,981$)) and type 2 diabetes ($n = 159,208$) and **B** their corresponding multivariable MR estimates. Odds ratios are per change in body size category. 95% CI 95% confidence interval. Central estimates are illustrated as circles which were filled when confidence intervals did not overlap with the null. The data underlying these figures can be found in Supplementary Data 9, 13 and 16.

MR analyses using our previously developed framework (with $n = 280$ genetic instruments).

2. Determine whether these childhood estimates based on age 10 years body size remain robust after accounting for very early life body size as proxied by genetically predicted birthweight.
3. Evaluate the converse relationships using MR i.e. whether T1D genetic liability influences body size in childhood or adulthood.
4. Investigate whether childhood body size has direct and indirect effects on seven other types of immune-associated or autoinflammatory diseases.

## Results
**Estimating the total effect of childhood body size on type 1 diabetes risk**. An overview of the exposure and outcome datasets used in the study can be found in Supplementary Data 1. Univariable MR analyses using the inverse variance weighted (IVW) method provided evidence that both childhood body size (Odds Ratio (OR) = 2.05 per change in body size category, 95% confidence interval (CI) = 1.20 to 3.50, $P = 0.008$) and adult body size (OR = 1.60, 95% CI = 1.05 to 2.45, $P = 0.03$) increase risk of T1D. The total effect of childhood body size was additionally supported by the MR-Egger method (OR = 5.06, 95% CI = 1.52 to 16.81, $P = 0.009$), suggesting that this result is robust to horizontal pleiotropy. In contrast, we obtained no convincing support that adult body size influences T1D based on the MR-Egger method (OR = 2.55, 95% CI = 0.72 to 9.00, $P = 0.145$) (Supplementary Data 2).

Repeating our univariable IVW analysis using genetic instruments for childhood and adult height provided no support of effects on T1D (childhood height: OR = 1.16, 95% CI = 0.94 to 1.44, $P = 0.174$, adult height: OR = 1.08, 95% CI = 0.85 to 1.36, $P = 0.532$) (Supplementary Data 3). These findings suggest that our estimates for childhood body size on T1D are capturing adiposity driven effect as opposed to a general body size effect. Furthermore, evidence of a total effect between childhood body size on T1D risk was validated using data from the largest available T1D meta-analysis to date (IVW: OR = 1.84, 95% CI = 1.19 to 2.83, $P = 0.006$, MR-Egger: OR = 3.28, 95% = 1.24 to 8.67, $P = 0.017$) (Supplementary Data 4). Age at diagnosis information for T1D from cohorts contributing to this meta-analysis can be found in Supplementary Data 5.

We also identified limited evidence of a reverse direction of effect between T1D genetic liability and childhood body size (Beta = 0.002 per 1-SD change in T1D liability, 95% CI = $-0.001$ to 0.005, $P = 0.236$), meaning that the effect of childhood body size on T1D is unlikely to be explained by reverse causality. Genetic instruments for T1D liability are reported in Supplementary Data 6. There was also little evidence to suggest that T1D genetic liability has an effect on BMI in adulthood (Beta = $-0.002$, 95% CI = $-0.005$ to 0.001, $P = 0.266$) (Supplementary Data 7). Conducting this reverse MR analysis using data from the Avon Longitudinal Study of Parents and Children (ALSPAC) supported these findings using measured childhood BMI at a mean age of 9.9 years in the life course (Beta = 0.033 per 1-SD change in T1D GRS, 95% CI = $-0.040$ to 0.106, $P = 0.382$). Investigating how our childhood and adult body size instruments relate to measured BMI in the ALSPAC cohort found that the childhood body size score associates more strongly with BMI, although not just using data measured from the mean age 9.9 years clinic but at 11 other earlier time points during childhood (Supplementary Fig. 2 and Supplementary Data 8).

**Evaluating the direct and indirect effects of childhood body size on type 1 diabetes risk**. Multivariable MR provided evidence that childhood body size has a direct effect on T1D risk (OR = 2.27, 95% CI = 1.24 to 4.17, $P = 0.008$), whereas adult estimates identified in this analysis included the null (OR = 0.92, 95% CI = 0.54 to 1.57, $P = 0.760$) (Fig. 2 and Supplementary Data 9). Using the multivariable MR-Egger method supported evidence of a direct effect for childhood body size on T1D risk (OR = 2.20, 95% CI = 1.20 to 4.05, $P = 0.011$) (Supplementary Data 10).

Repeating our multivariable MR analyses on T1D risk with the addition of genetically predicted birthweight in the model found that the childhood body size estimates were maintained (OR = 2.32, 95% CI = 1.21 to 4.42, $P = 0.013$) (Supplementary Data 11). Additionally, higher genetically predicted birthweight provided evidence of a protective direct effect on T1D risk (OR = 0.58, 95% CI = 0.41 to 0.82, $P = 0.002$) independent of childhood and adult body size. These results suggest that body size at birth is unlikely to be responsible for the effect of childhood body size on T1D in our model. Furthermore, univariable estimates for birthweight on T1D risk were not robust to horizontal pleiotropy based on estimates from the MR-Egger method (OR = 0.44, 95% CI = 0.16 to 1.24, $P = 0.124$) (Supplementary Data 12). Multivariable MR estimates for adult body size on T1D, accounting for genetically

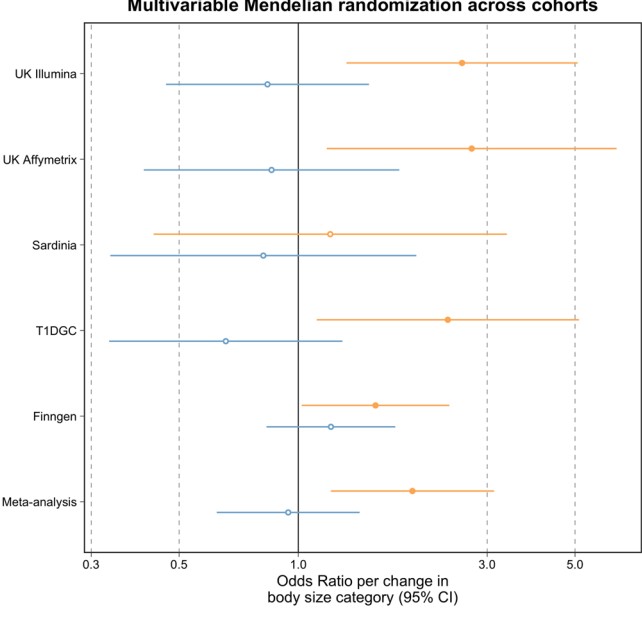

**Multivariable Mendelian randomization across cohorts**

Odds Ratio per change in
body size category (95% CI)

● Childhood body size  ● Adult body size

**Fig. 3 A forest plot depicting multivariable Mendelian randomisation estimates of childhood body size on type 1 diabetes risk for each study contributing to the meta-analysis.** Multivariable Mendelian randomisation analyses of childhood (yellow) and adult (blue) body size (n = 453,169) on type 1 diabetes risk were undertaken in each contributing study to the large-scale meta-analysis used in this work (n = 173,981). Odds ratios are per change in body size category. 95% CI 95% confidence interval. Central estimates are illustrated as circles which were filled when confidence intervals did not overlap with the null. T1DGC Type 1 Diabetes Genetics Consortium. The data underlying this figure can be found in Supplementary Data 14.

predicted birthweight, did not support a role of obesity later in life influencing T1D (OR = 0.77, 95% CI = 0.43 to 1.39, P = 0.390).

Evidence of a direct effect between childhood body size and T1D risk was validated using data from the large meta-analysis of T1D GWAS (OR = 1.94, 95% CI = 1.21 to 3.12, P = 0.006) (Supplementary Data 13). Direct effect estimates derived from each contributing dataset to the T1D meta-analysis were typically consistent with the exception of the cohort from Sardinia (Fig. 3, Supplementary Figs. 3, 4 and Supplementary Data 14). Assuming that our OR estimates are approximately equal to relative risks (RRs), and assuming a T1D prevalence of 0.5%, we used our direct childhood body size MR-Egger OR estimate from the meta-analysis (OR = 2.64) to build a table of proportions for T1D affected and unaffected individuals lying in each body size category. Mimicking an intervention and assuming a constant RR, we reduced the proportion of individuals who are 'plumper than average' from 0.159 (i.e. the proportion within our data) to 0.059, and increased the proportion in the 'slimmer than average' category from 0.33 to 0.43. This was to reflect a simplified but realistic scenario in which 10% of individuals move from the high weight to the average weight category, and the same number of average weight individuals move into the low weight category. Our intervention model produced a fall in T1D prevalence to 0.39% (a 22% reduction) (Supplementary Data 15).

We also repeated our multivariable MR analysis of childhood and adult body size with T2D as an outcome to generate revised estimates compared to our previous work[13]. In contrast to our results for T1D, these estimates suggest that childhood body size has an indirect on T2D as our univariable childhood estimates (OR = 2.18, 95% CI = 1.80 to 2.63, P = 8.91 × 10$^{-16}$) were

reduced and included the null when accounting for adult body size (OR = 0.90, 95% CI = 0.69 to 1.19, P = 0.465) (Supplementary Data 16).

**Investigating whether childhood body size directly influences other types of immune disease.** We applied univariable and multivariable MR analyses to investigate the total, direct and indirect effects of childhood body size on seven other immune-associated diseases in turn: asthma, atopic dermatitis and eczema, hypothyroidism, rheumatoid arthritis, inflammatory bowel disease and its two subtypes (Crohn's disease and ulcerative colitis) (Supplementary Data 1). Using univariable MR, 9 of the 14 analyses undertaken provided evidence that body size in either childhood or adulthood influenced chronic immune disease risk based on FDR <5% (Supplementary Data 17). For childhood body size, this included evidence of increased asthma risk (OR = 1.31, 95% CI = 1.08 to 1.60, P = 0.007), dermatitis and eczema (OR = 1.25, 95% CI = 1.03 to 1.51, P = 0.024) and hypothyroidism (OR = 1.42, 95% CI = 1.12 to 1.80, P = 0.004). Adult body size provided evidence of influencing risk on outcomes including Crohn's disease (OR = 1.37, 95% CI = 1.10 to 1.70, P = 0.005) and rheumatoid arthritis (OR = 1.42, 95% CI = 1.05 to 1.93, P = 0.022).

Using multivariable MR, the direct effect estimates for childhood body size on all immune-associated outcomes which provided evidence of an effect in a univariable setting included the null when accounting for the effect of adult body size (Supplementary Data 17). There was stronger evidence however that childhood body size indirectly influences disease risk via adult body size on; asthma risk (OR = 1.30, 95% CI = 1.04 to 1.63, P = 0.022), dermatitis and eczema (OR = 1.30, 95% CI = 1.03 to 1.64, P = 0.026) and hypothyroidism (OR = 1.94, 95% CI = 1.45 to 2.61, P = 9.64 × 10$^{-6}$). After correcting multivariable analyses for false discovery rates (FDR), only the effect on hypothyroidism remained robust (FDR = 1.35 × 10$^{-4}$). All univariable and multivariable MR estimates derived in these analyses have been illustrated using forest plots in Fig. 4.

## Discussion

We present evidence suggesting that body size in childhood increases the risk of T1D based on the age-at-diagnosis of the participants analysed in this study (mean age = 16.57 years). These findings support previous results from observational studies suggesting that the increasing prevalence of childhood obesity is a causal factor in the rising numbers of T1D diagnoses. Systematically applying our MR framework to seven other immune-associated diseases suggested, initially, that childhood body size also increases the risk of asthma, eczema and hypothyroidism. However, these effect estimates attenuated once accounting for adulthood body size, suggesting that they can be explained due to the sustained impact of adiposity among children who are overweight and thus tend to remain so into adulthood.

The effect of genetically predicted childhood body size on T1D risk could have various explanations. For instance, this evidence may support findings from the literature suggesting that excess fat tissue has a deleterious influence on the body's immune system, potentially with secreted adipokines playing a mediatory role[20]. As outlined by the 'accelerator hypothesis'[21], increased stress on insulin demands in children with obesity may contribute to earlier β-cell failure and subsequently an earlier diagnosis of T1D[22]. Evidence from a mouse model of non-immune diabetes induced by a high-fat diet indicated that diabetes can result from β-cell fragility[23], including genetically lower expression of the transcription factor gene, *GLIS3*, which is known to be associated

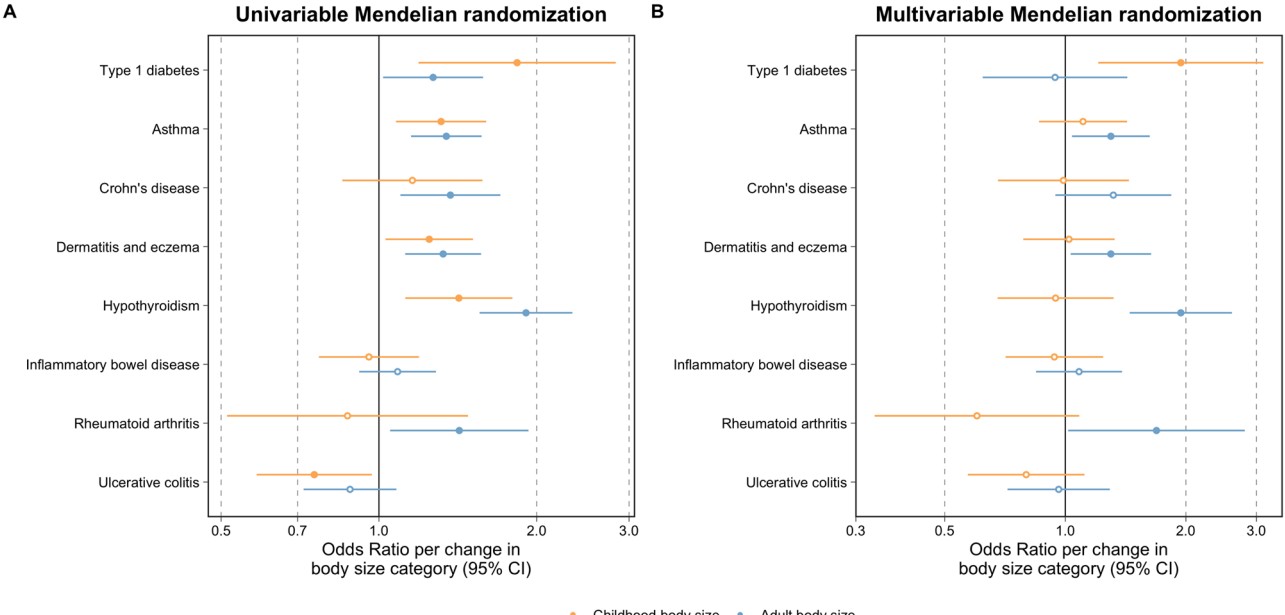

**Fig. 4 Forest plots comparing the univariable and multivariable Mendelian randomisation estimates of childhood body size on type 1 diabetes risk and seven chronic immune disease outcomes. A** The univariable Mendelian randomisation (MR) estimates between childhood (yellow) and adult (blue) body size ($n = 453,169$) on the risk of chronic immune disease outcomes (see Supplementary Data 1 for sample sizes) and **B** their corresponding multivariable MR estimates. The type 1 diabetes estimates were based on the analysis using data from Crouch et al. (2021) ($n = 173,981$). Odds ratios are per change in body size category. 95% CI 95% confidence interval. Central estimates are illustrated as circles which were filled when confidence intervals did not overlap with the null. The data underlying this figure can be found in Supplementary Data 13 and 17. Multiple testing comparisons for estimates in this figure were accounted for by calculating false discovery rates as reported in Supplementary Data 17.

with susceptibility to both T1D and T2D[24]. High fat and carbohydrate diets with low fibre in early life, resulting in childhood obesity, could compromise the metabolic and immune functions of the gut microbiome, where microbiota dysbiosis has been associated with both T2D[25] and T1D[26]. Regardless of the underlying mechanisms, our findings suggest that a critical window exists in childhood to mitigate the influence of adiposity on the escalating numbers of T1D diagnoses and that an ~22% reduction in the number of T1D cases is plausible if the proportion of children within the highest obesity category were to be reduced by 10%, from 15.9 to 5.9%.

As expected given the average age-at-diagnosis of T1D, the effect of childhood body size remained robust after accounting for adult body size using a much larger number of genetic instruments than previously used ($n = 280$ in this study versus $n = 23$ previously[17]). Furthermore, our childhood estimates remained strong even after accounting for birthweight. However, estimates derived using the MR-Egger method only supported the childhood body size effect (OR = 5.06, 95% CI = 1.52 to 16.81, $P = 0.009$), whereas confidence intervals for both birthweight and adult body size overlapped with the null meaning that they were not as strongly supported by this approach of having a genetically predicted effect on T1D risk.

In particular, the multivariable MR estimates for adult body size illustrate the importance of using our approach to separate the effects of body size at separate stages in the life course. This is because the univariable MR estimates for adult body size on their own could conceivably be interpreted as evidence that it influences T1D risk, which is unlikely given the age of onset for this disease in the study sample analysed. However, taken together with the MR-Egger estimates, our multivariable analysis suggested that adult body size does not influence T1D risk after accounting for the effect of childhood body size. Further work is required to investigate late-onset T1D using age-at-diagnosis data

once it becomes available in large sample sizes, particularly given the challenges of T1D diagnosis in adulthood[27]. This would be valuable as it would facilitate investigation into whether adiposity in adulthood increases the risk of late-onset T1D, which our study may be underpowered to detect due to the large majority of individuals in our T1D sample being diagnosed during childhood.

We incorporated birthweight as an additional exposure in our multivariable model to assess whether it may help explain the effect of childhood body size on T1D. As our estimates remained robust, these findings do not seem to suggest that variation in birthweight is responsible for the effect of genetically predicted childhood body size on T1D risk identified in our analysis. However, a more appropriate evaluation of the influence of birthweight on T1D risk requires in-depth evaluation using both maternal and foetal genetic effects, as undertaken previously, once sample sizes of both maternal and offspring T1D cases are sufficient[28,29]. Amongst other sources of bias, future endeavours applying this study design will be able to investigate whether our results may be underestimating the genetically predicted effect of birthweight on T1D risk.

Our MR analysis on other types of immune-associated disease suggested that the childhood body size effect on T1D is not generalisable to other chronic immune outcomes. Amongst this finding was evidence of a total effect of childhood body size on later life asthma risk which corroborated recent MR results suggesting that increased asthma risk is likely explained by individuals remaining overweight into adulthood[30]. However, our univariable results provide stronger evidence than previously reported that the effect of adiposity on asthma risk begins in childhood, which may potentially be explained by the influence of excess abdominal fat driving systemic inflammation[31]. In particular, our findings suggest that adiposity begins to exert its effect on the risk of eczema and hypothyroidism in childhood, which has previously been reported in the literature by non-genetic

studies[32,33]. The attenuation of the childhood estimates on these outcomes in our multivariable model suggests that adiposity influences their risk due to a sustained effect of remaining overweight for many years across the life course (similar to our findings for T2D[34]). Further research is therefore necessary to verify whether lifestyle changes enforced post-childhood can alleviate the detrimental effect of childhood body size on these outcomes as with T2D[12]. Furthermore, if this is the case then extensive research into the critical windows where this effect begins to become immutable will be extremely important to identify for disease prevention purposes.

There are various strengths and limitations of our study which should be taken into account when interpreting its findings. Firstly, the use of genetic variation in a two-sample MR framework allowed us to analyse a large number of genetic instruments from the UK Biobank sample for body size ($n = 454,023$) with a meta-analysed sample of T1D cases (up to $n = 15,573$), almost twice the number of cases used in a previous study[17]. As such our results are less prone to bias attributed to reverse causation and confounding factors compared to more traditional epidemiology approaches. Furthermore, this study design allowed us to investigate the direct and indirect effects of childhood body size on T1D as well as seven other chronic immune-associated diseases in turn, which would be extremely challenging to undertake without the use of human genetics. Conversely, one of the major limitations of this work is that our 280 genetic instruments for childhood body size were derived using recall data which may be more prone to bias due to factors such as measurement error. That said, previously conducted simulations and extensive validation studies in three separate populations[13,35,36] support the use of these instruments to separate the effect of childhood body size using these instruments from that of adulthood body size. A further limitation of our study is that childhood body size was measured at age 10, while 47.9% of our T1D meta-analysis cases were diagnosed before 10. Among this subset of cases, we cannot eliminate the possibility that exposure to obesity occurred after developing T1D, which would preclude a causal relationship. However, we believe our study to have good statistical power due to (a) 49.5% of cases having known age-at-diagnosis older than 10 and (b) obesity at age 10 being presumably correlated with obesity at earlier ages, providing effective exposure prior to disease diagnosis for a larger subset of T1D cases. FinnGen effect estimates were similar to most other cohorts despite containing a far greater proportion of cases diagnosed over age 20 (62%), supporting (b). Likewise, evaluations of our genetic instrument for childhood body size found that it is strongly associated with measured BMI in the ALSPAC cohort throughout early life and not just at age 10 (as depicted in Supplementary Fig. 2).

Additionally, although MR studies are typically considered to be less prone to reverse causation than observational studies, there are possible scenarios where this could still bias findings as outlined in a recent review[37]. This is why in this study we investigated the converse direction of effect for our primary analysis using MR i.e. whether T1D genetic liability influences childhood body size. As weak evidence of an effect was found in this sensitivity analyses, our findings suggest that T1D resides downstream of childhood BMI and also that a scenario involving feedback mechanisms are unlikely. Accounting for birthweight in our model also mitigates the likelihood that a cross-generational effect is underlying the genetically predicted effect of childhood body size on T1D risk found in our study.

In conclusion, our findings emphasise the importance of implementing preventative policies to lower the prevalence of childhood obesity and its subsequent influence on the rising numbers of T1D cases. This will help ease healthcare burdens and also potentially improve the quality of life for individuals living with this lifelong disease.

## Methods

**Data resources.** All individual participant data used from the UK Biobank (UKB) study had ethical approval from the Research Ethics Committee (REC; approval number: 11/NW/0382) and informed consent from all participants enroled in UKB. Ethical approval for data obtained from the Avon Longitudinal Study of Parents and Children (ALSPAC) was obtained from the ALSPAC Ethics and Law Committee and the Local Research Ethics Committees.

**Genetic instruments for childhood and adult body size.** Genetic variants associated with childhood and adult body size (based on $P < 5 \times 10^{-8}$) were identified from a previously undertaken GWAS in the UKB study[38,39]. In these GWAS we derived our childhood body size measure using recall questionnaire data asking UKB participants if they were 'thinner', 'plumper' or 'about average' when they were aged 10 years old compared to the average. Adult body size was derived using clinically measured body mass index (BMI) data (mean age 56.5 years), which we categorised into a three-tier variable using the same proportion as the early life measure for comparative purposes.

GWAS were undertaken on 453,169 individuals who had both measures available with adjustment for age, sex and genotyping chip. Our GWAS of childhood body size was additionally adjusted for month of birth. We used a linear mixed model to account for genetic relatedness and geographical structure in UKB as undertaken with the BOLT-LMM software[40]. In the original study where these instruments were derived we did not identify any evidence against a linear relationship between our exposure variable in line with the assumptions of multivariable MR[13]. To support the robustness of these instruments in terms of their ability to separate the effects of childhood and adult body size, we have previously undertaken validation analyses using measured BMI data from three independent populations: the ALSPAC study[13], the Young Finns Study[41] and the Trøndelag Health (HUNT) study[36]. Other validation analyses have also been conducted previously, whereby GWAS results for the childhood measure had a higher genetic correlation with measured childhood obesity from an independent sample (rG = 0.85) compared to the adult measure (rG = 0.67). Conversely, genome-wide estimates for the adult measure were more strongly correlated with measured BMI in adulthood (rG = 0.96) compared to the childhood measure (rG = 0.64)[13]. Furthermore, using these instruments previously for multivariable MR provided F-statistics >10 suggesting that derived results are unlikely to be prone to weak instrument bias[13].

**Genetic instruments for childhood height, adult height and birthweight.** For this study, we repeated the same protocol described above but for childhood and adult height using data from the UKB study, to demonstrate that our body size was likely capturing adiposity rather than being bigger at age 10. Participants were asked "When you were 10 years old, compared to average would you describe yourself as…", and given the options of 'shorter', 'about average' or 'taller'. GWAS were undertaken as above on the childhood measure of height as well as a three-tiered categorical variable for adult measured height based on the same proportions. GWAS on childhood and adult height were undertaken on 454,023 individuals who had both measures available with adjustment for the same covariates as before. The same analysis pipeline was applied to generate genetic instruments for birthweight which was kept continuous due to only being available on a total of 261,932 UKB individuals. This trait was rank-based inverse normal transformed to ensure normality and adjusted as before for age, sex and genotyping chip.

**Genetic effects on T1D, T2D and other immune-associated diseases.** Genetic estimates for all outcomes analysed in this study were obtained from large-scale GWAS studies and meta-analyses conducted by consortia. We first applied our multivariable approach using a large number of childhood and adult body size instruments to T1D data analysed previously in the study by Censin et al. ($n = 5913$ cases diagnosed before the age of 17 years and $n = 8828$ controls). Results from this analysis were then validated using a recent large-scale meta-analysis of up to 15,573 cases and 158,408 controls[42]. Analyses were then repeated separately in each contributing cohort from this meta-analysis: Illumina genotyped UK samples (3983 cases and 3994 controls), Affymetrix genotyped UK samples (1926 cases and 3342 controls), Sardinians (1558 cases and 2882 controls), Finnish FinnGen samples (4933 cases and 148,190 controls) and the Type 1 Diabetes Genetics Consortium (T1DGC) European-ancestry family sample (3173 affected-offspring trios, analysed by the transmission disequilibrium test).

In terms of age-at-diagnosis, 7,453 T1D meta-analysis cases were diagnosed before 10 years of age, 4368 between 10 and 20 years old, 3352 over 20 years old and 400 with missing data) (see Supplementary Data 5 for a breakdown by cohort). As such nearly half of the cases included in the meta-analysis had age-at-diagnosis later than 10, the age at which our childhood body size instrument is based on.

We also obtained estimates using results from a GWAS of T2D, updated since our previous study[43], and seven of the most common immune-associated disease

endpoints: asthma, atopic dermatitis and eczema, hypothyroidism, rheumatoid arthritis, inflammatory bowel disease and it's two subtypes (Crohn's disease and ulcerative colitis). An overview of these outcome datasets and all others analysed in this study can be found in Supplementary Data 2.

**Instrument identification and data harmonisation**. We previously constructed a reference panel-based using genotype data from 10,000 unrelated UK Biobank participants of European descent to undertake linkage disequilibrium (LD) clumping[44]. This allowed us to identify independent genetic variants for MR analyses based on an LD cutoff of $r^2 < 0.001$[45], which was necessary to ensure MR estimates were not biased by using correlated instruments. For multivariable MR, we repeated LD clumping but used aggregated sets of genetic variants for all our exposures to ensure they were also independent. Genetic estimates for our exposures were harmonised with disease outcomes using the 'TwoSampleMR' R package[46]. In total, there were 280 childhood body size and 515 adult body size instruments available for analysis after harmonisation with T1D genetic estimates, where 81 were subsequently removed prior to conducting multivariable MR analyses. Additionally, 629 childhood height and 907 adult height instruments and 161 birthweight instruments were identified for sensitivity analyses. The number of instruments for all subsequent analyses varied depending on factors such as coverage, population allele frequencies and the strand alignment of corresponding GWAS results.

**The Avon Longitudinal Study of Parents and Children**. ALSPAC is a population-based cohort investigating genetic and environmental factors that affect the health and development of children. The study methods are described in detail elsewhere[47,48]. In brief, 14,541 pregnant women residents in the former region of Avon, UK, with an expected delivery date between April 1, 1991 and December 31, 1992, were eligible to take part in ALSPAC. Detailed phenotypic information, biological samples and genetic data which have been collected from the ALSPAC participants are available through a searchable data dictionary and variable search tool (http://www.bris.ac.uk/alspac/researchers/our-data/). Consent for biological samples has been collected in accordance with the Human Tissue Act (2004). Written informed consent was obtained for all study participants. Ethical approval for this study was obtained from the ALSPAC Ethics and Law Committee and the Local Research Ethics Committees.

In ALSPAC, height was measured to the nearest 0.1 cm with a Harpenden stadiometer (Holtain Crosswell), and weight was measured to the nearest 0.1 kg on Tanita electronic scales to derive measures of BMI (weight (kg)/height (m)$^2$). BMI measures were collected at multiple time points across childhood, including ten measures from age 4 months old to 5 years old, as well as at the Focus at age 7 clinic (mean age = 7.5 years, range = 7.1 years to 8.8 years) and the Focus at age 9 clinic (mean age = 9.9 years, 8.9 years to 11.5 years). A summary of these measures can be found in Supplementary Data 2.

**Statistical analysis**
*Univariable Mendelian randomisation.* We firstly undertook univariable MR analyses to evaluate the total effect of genetically predicted childhood body size on T1D risk. We applied the inverse variance weighted (IVW) method for initial analyses, which takes the SNP-outcome estimates and regresses them on those for the SNP-exposure associations[49]. The weighted median and MR-Egger methods were subsequently applied as sensitivity analyses to evaluate the robustness of IVW estimates to horizontal pleiotropy[50,51]. This is the phenomenon whereby genetic variants influence exposure and outcome via two separate biological pathways[15].

Univariable analyses with T1D as an outcome were repeated separately for adult body size and for birthweight. We included adult body size to demonstrate the importance of using genetic scores to separate the effects of body size at different stages in the life course when investigating either early or late-onset disease outcomes. Additionally, we investigated the opposite direction of effect using the same univariable methods mentioned above to assess whether genetic liability towards T1D risk influences body size in both childhood and adulthood in turn. In this analysis we used a set of 63 genetic instruments for T1D identified from a recent meta-analysis (of up to 15,573 cases and 158,408 controls[42]) which had an F-statistic of 196 (Supplementary Data 6). Non-HLA SNPs in this score were selected to be independent (pairwise $r^2 < 0.01$), while the 5 HLA SNPs had pairwise $r^2 < 0.02$ in European non-Finnish 1000 Genomes samples (using the LDlink online tool, https://ldlink.nci.nih.gov/). Adult BMI was analysed as a continuous trait to derive a per standard deviation effect estimate. Additionally, we derived a genetic risk score (GRS) using data from the ALSPAC cohort and investigated the effect of T1D genetic liability on measured BMI data from the mean age of 9.9 years (range = 8.9 to 11.5 years old) clinic adjusting for age and sex. We also used data from the ALSPAC cohort to evaluate how our childhood and adult body size instruments relate to BMI at 12 time points in childhood prior to age 10.

Birthweight was analysed in this study to investigate whether an individual's body size in very early life (e.g. before age 5 years) may be responsible for the effects identified using our childhood genetic score (Supplementary Fig. 1). These analyses were not however intended as an exploration of the effects of parental influences on T1D risk[52], as birthweight variation is known to be influenced by a combination of both foetal and parental genetic and non-genetic factors[28]. We also repeated analyses on T1D using instruments for childhood and adult height to demonstrate that our childhood body size measure was capturing childhood body size (i.e. being 'plumper' as described in the questionnaire) rather than being taller than the other 10-year olds.

*Multivariable Mendelian randomisation.* We next sought to estimate the direct and indirect effect of childhood body size on T1D risk using multivariable IVW MR[53,54]. This was firstly undertaken by accounting for adult body size as an additional exposure in our model (i.e. alongside childhood body size), and subsequently including birthweight as a third exposure. We also applied the multivariable MR-Egger method to evaluate horizontal pleiotropy for the direct and indirect effects of childhood body size[55]. Furthermore, multivariable analyses were repeated using data from the large-scale T1D meta-analysis[42], as well as evaluating evidence using data from each contributing cohort to this dataset in turn. Lastly, we repeated our multivariable MR analysis with childhood and adult body size as exposures to each of the seven different types of immune-associated/autoinflammatory disease in turn. To account for multiple testing in this analysis, we applied the Benjamini-Hochberg false discovery rate (FDR) correction of FDR <5%.

Forest plots in this paper were generated using the R package 'ggplot2'[56]. All analyses were undertaken using R (version 3.5.3).

**Reporting summary**. Further information on research design is available in the Nature Research Reporting Summary linked to this article.

## Data availability
All individual-level data analysed in this study can be accessed via an approved application to ALSPAC (http://www.bristol.ac.uk/alspac/researchers/access/). Summary statistics on type 1 and type 2 diabetes are publicly available from the studies as referenced in Supplementary Data 2. All other summary statistics analysed in this study can be accessed via the OpenGWAS (https://gwas.mrcieu.ac.uk/) and FinnGen (https://www.finngen.fi/fi) resources.

## Code availability
Univariable and multivariable MR analyses were conducted using the TwoSampleMR package (version 0.5.5) in R (version 3.5.3).

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

## Acknowledgements

We thank the authors and GWAS consortia who made their summary statistics available for the benefit of this study. This work was supported by the Integrative Epidemiology Unit which receives funding from the UK Medical Research Council and the University of Bristol (MC_UU_00011/1). G.D.S. conducts research at the NIHR Biomedical Research Centre at the University Hospitals Bristol NHS Foundation Trust and the University of Bristol. The views expressed in this publication are those of the author(s) and not necessarily those of the NHS, the National Institute for Health Research or the Department of Health. TGR was a UKRI Innovation Research Fellow (MR/S003886/1) whilst undertaking this project. G.M.P. is supported by grant MR/N0137941/1 for the GW4 Biomed Doctoral Training Programme, awarded to the Universities of Bath, Bristol, Cardiff and Exeter from the Medical Research Council (MRC)/UKRI. S.F. and F.M.-B. are supported by Wellcome Trust PhD studentships in Molecular, Genetic and Lifecourse Epidemiology [108902/Z/15/Z and 218495/Z/ 19/Z, respectively]. We are extremely grateful to all the families who took part in this study, the midwives for their help in recruiting them and the whole ALSPAC team, which includes interviewers, computer and laboratory technicians, clerical workers, research scientists, volunteers, managers, receptionists and nurses. The UK Medical Research Council and Wellcome (Grant ref: 217065/Z/19/Z) and the University of Bristol provide core support for ALSPAC. Genetic data were generated by Sample Logistics and Genotyping Facilities at the Wellcome Trust Sanger Institute and LabCorp (Laboratory Corporation of America) using support from 23andMe. The work of D.J.M.C., J.R.J.I. and J.A.T. was supported by the JDRF [9-2011-253], [5-SRA-2015-130-A-N], [4-SRA-2017-473-A-N]; the Wellcome [091157/Z/10/Z], [107212/Z/15/Z]; [203141/Z/16/Z]. No funding bodies had any role in study design, data collection and analysis, decision to publish, or preparation of the manuscript. Computation used the Oxford Biomedical Research Computing (BMRC) facility, a joint development between the Wellcome Centre for Human Genetics and the Big Data Institute supported by Health Data Research UK and the NIHR Oxford Biomedical Research Centre. Financial support was provided by the Wellcome Trust Core Award Grant Number 203141/Z/16/Z. The views expressed are those of the author(s) and not necessarily those of the NHS, the NIHR or the Department of Health.

## Author contributions

G.D.S. conceived the initial study design. T.G.R. carried out data curation and primary analyses as well as validation analyses in the ALSPAC cohort. D.J.M.C. conducted validation analyses using the large type 1 diabetes meta-analysis. Type 1 diabetes summary genetic data were obtained through multiple collaborations from studies led by D.J.M.C., J.R.J.I., C.C.R., C.S., F.C., S.S.R. and J.A.T. Additional analyses and the generation of supporting figures were performed by G.M.P., F.M.-B., E.H., S.F. and Y.C. The initial manuscript was drafted by T.G.R., D.J.M.C., J.A.T. and G.D.S. All authors (T.G.R., D.J.M.C., G.M.P., F.M.-B., E.H., S.F., Y.C., J.R.J.I., C.C.R., C.S., F.C., S.S.R., J.A.T. and G.D.S.) contributed to the interpretation of the results and critical revision of the manuscript. This work was jointly supervised by J.A.T. and G.D.S.

## Competing interests

J.A.T. is a member of the Human Genetics Advisory Board of GSK. T.G.R. is employed part-time by Novo Nordisk outside of this work. The remaining authors declare no competing interests.
