## [Peer Review File · Nature Communications]

Childhood body size directly increases type 1 diabetes risk based on a lifecourse Mendelian randomization approachReviewers' Comments:

Reviewer #1:

Remarks to the Author:

Richardson et al present a Mendelian Randomisation analysis to estimate the relative causal effect sizes of childhood and adult BMI on Type 1 Diabetes as well as a number of other autoimmune diseases. This is an well written and interesting study addressing an important question, finding a causal role for childhood BMI on T1D, while the role of adult BMI was more important for the majority of the other diseases tested. I have some minor comments listed below:

1. The authors used UK Biobank to select genetic variants to use as instrumental variables for adult BMI by categorising the raw continuous BMI variable to be consistent with the childhood BMI instruments. The authors also used UK Biobank to select instruments for birth weight, but I was unclear as to whether the birthweight phenotype was categorised in the same way as the adult BMI phenotype. The methods suggest that it was not. Could the authors clarify whether the birthweight phenotype was categorised or not, and if not why was the decision taken to categorise adult BMI but not birthweight?
2. Results line 261: The authors write "Beta=-0.007 per standard deviation change in BMI" which suggests that this result is from an MR of the effect of BMI on T1D but the text suggests that this result is from an MR of the effect of T1D on BMI – should the units not be SD change in BMI per log odds T1D liability?
3. In the supplemental tables it was not immediately clear to be whether effect size estimates given were odds ratios or log-odds ratios – these were often just labelled "Beta" or "estimate". Could the authors clarify in table captions what units the figures given relate to?
4. Line 313: "Using univariable MR, 10 of the 16 analyses undertaken provided evidence that adiposity in either childhood or adulthood influenced chronic immune disease risk based on FDR<5% (Supplementary Table 13)" – Table S13 seems to only include 14 analyses (7 traits, each for childhood and adulthood BMI)
5. Could the authors comment on how confident they are that the estimates of childhood and adult BMI effects on for example Asthma. While the effect estimate for childhood BMI includes the null but the adult BMI estimate does not, the effect estimates are similar, and their confidence intervals include each other. Does this affect the interpretation that childhood BMI only has indirect effects on asthma?
6. In the discussion, line 375 the authors state that the fact that the MR-Egger 95% confidence interval for the adult BMI and birthweight effects on T1D overlap the null suggest evidence of "horizontal pleiotropy". It is my understanding that the intercept of the MR-Egger analysis is a measure of the presence of unbalanced horizontal pleiotropy, and that MR-Egger is known to be less powerful than IVW due to the additional degree of freedom in the estimation of both the slope and intercept parameters. Is the interpretation that the MR-Egger estimate includes the null indicates the presence of pleiotropy correct?
7. The authors include birthweight in their multivariable MR analysis. They state that the reason for this is to investigate whether individual's body size in very early life may be responsible for the effects identified using childhood BMI instrument. While they acknowledge that their analyses do not account for maternal effects, they give little discussion to how much impact this could have on their conclusions. For instance, the correlation between maternal and fetal genotype can lead to both under- and over-estimation of SNP-exposure and SNP-outcome associations. If these over- or under-estimations result in an under-estimation of the causal effect size of birthweight on T1D this could lead to an incorrect conclusion that birthweight does not causally impact on T1D risk but childhood BMI does. As such, I would like to at least see some consideration of the impact of this on the results in the discussion, and I also feel that the statement in the first line of the discussion "We present evidence that adiposity in childhood increases the risk of T1D independently of body size at birth and adulthood" should be toned down in relation to size at birth.
8. The authors mention that adult BMI is unlikely to causally impact T1D risk give the age of onset of this disease, however T1D does not occur only in childhood and can manifest only in later life. Could

the authors comment in the discussion of whether their results suggest that childhood BMI a more important causal factor than adult BMI in later onset T1D, or whether the fact that the majority of cases in the T1D GWAS would have had early onset mean that these results are not generalisable to later onset T1D?

Reviewer #2:

Remarks to the Author:

The study uses several very large data sets to conduct state of the art mendelian randomization (MR) analyses attempting to disentangle the potential direct and indirect causal effects of childhood and adult adiposity on type 1 diabetes T1D and other immune-mediated diseases such as asthma, inflammatory bowel diseases (and extending previous results on type 2 diabetes). The main conclusions are that adiposity in childhood rather than in adulthood is directly causally related to the risk of type 1 diabetes, while the opposite seems to be the case for T2D, asthma, eczema and hypothyroidism. The T1D result confirms and extends a previous finding by Censin using univariate MR analyses (ref 15 in the manuscript).

The authors are top experts and analyses seems largely well done, and the paper contributes some novel results. My most important concern is that I struggle with the idea that adult adiposity should mediate the effect of childhood adiposity, or indeed have any causal effect on diseases with typical onset in childhood, be it T1D, asthma or eczema. The authors could do a better job at explaining to readers how this makes sense. I have some additional comments as well as a number of mostly minor corrections or suggestions. (some of the authors have recently contributed several other interesting papers using similar approaches for type 2 diabetes and other disease outcomes typically occurring in adulthood, where it makes more sense to me).

COMMENTS

1.Role of adult BMI in early onset disease: First let me say that although I have a decent understanding of traditional mediation analyses and simple MR analyses, I have no first hand analysis experience with multivariable MR (MVMR) for mediation analysis. My comments may reflect this, but I believe this is also likely to be the case for 99% of readers of this paper.

a.On line 368-370, the authors write: "As expected given the average age-at-diagnosis of T1D, the effect of childhood body size remained robust after accounting for adult body size using a much larger number of genetic instruments than previously used (n=280 in this study versus n=13 previously)". In a way, this states that the main research question is not interesting, as it is obvious that adult BMI cannot influence a disease which typically starts and is diagnosed in childhood. I believe this argument can be put forward also for asthma, eczema and perhaps some of the other immune-mediated diseases outcomes studied. Please explain. Is there something with the MR study that should make us think differently about how to interpret such mediation?

b.Information on age at diagnosis: I believe most readers would appreciate a table of characteristics which at least contains the mean and range of age at diagnosis. Some details on diagnostic criteria should be provided, e.g. if T1D in adults how was diabetes classified. Any diagnoses such as asthma based on self- or parent report?

c. Analyses by age at diagnosis: Under statistical analysis (line 200), the authors wrote: "We included adult body size to demonstrate the importance of using genetic scores to separate the effects of adiposity at different stages in the life course when investigating either early or late onset disease outcomes". I found no analysis of early vs late onset of a single disease, nor any explanation which of the specific diseases were considered early or late onset. At least for type 1 diabetes and the most important contributor to the genetic liability, HLA, there is evidence that the strength of association

varies by age at onset. Analyses by age at diagnosis of T1D and perhaps other diseases would probably be informative.

d. Given the above, I think the analysis accounting simultaneously for birth weight and childhood adiposity makes more sense. I appreciate the authors comment that BWT was not appropriately handled to include maternal effects but I think results similar to those done for T1D should be done also for the other disease outcomes (including T2D). My hunch is that although there may be maternal effects influencing birth weight, the fetal genes (inherited maternal and paternal) are likely most important in such a mediation analysis. Also, interpretation of the apparent inverse association of birth weight with T1D is difficult given previous observational studies suggesting weak but very consistent positive association across many large studies.

2. Reverse causation:

a. Effect of T1D genetic liability. Please provide more information on the GRS (in addition to citing a reference). How strongly does it predict T1D (eg. AUC), how were HLA SNPs handled – haplotypes, interactions etc?

b. On line 382, the authors write: “Whilst we did not find evidence that genetic liability towards T1D may influence childhood adiposity, your results suggest that it may have an effect on lower body size in adulthood based on the MR-Egger and weighted median methods. Medical practitioners promote healthy living among T1D patients in order to keep HbA1c levels low, which is one possible explanation for this result.” This is relevant for the general interpretation of MR results and mediation. My take on the author’s interpretation is that they believe the causal effect to be mediated via diagnosis of T1D and treatment of the disease (besides the fact that intensive insulin treatment to lower HbA1c will in fact tend to increase BMI, not lower it). I am wondering if this is a plausible interpretation at all, because in the general population, the incidence and prevalence of T1D is low (<1%), and if most or all of the effect of a T1D genetic risk score (GRS) was really mediated via diagnosis and treatment of T1D, I believe that this would not be detectable as a “total effect” in the population, given the low prevalence of the mediator. Unless I have misunderstood how these MR analyses should be interpreted, please rephrase this part of the discussion, and provide a more plausible explanation of this finding.

c. Estimating potential causal effect of T1D GRS on BMI in childhood: These analyses could preferably have been done in cohorts with measured childhood BMI (HUNT, ALSPAC, YFS, studies some of the authors have used previously for similar analyses of other outcomes), rather than the UKB where childhood adiposity around the age of 10 years was recalled (from mean age around 56 years at study enrolment) and categorised in three general categories, which is a weakness (both potential recall bias and coarse categories).

d. Results for “converse” effect: T1D GRS on adult BMI was reported as -0.007 per change in SD of BMI. This must be an error, as the analysis is change in BMI (presumably in units of SD, or what?) per change in (units?) of T1D GRS! Please correct and clarify.

e. Please also discuss potential scenarios outlined by Burgess and Swanson (Int J Epidemiol 2020) where MR analyses can be influenced by reverse causation type of biases - relevance for the current results?

3. Assumptions and interpretation of MVMR mediation analyses :

a. uncorrelated instruments: In traditional mediation analysis, the exposure and mediator can be correlated, whereas MVMR mediation analysis apparently requires uncorrelated instruments for childhood and adult adiposity. I would expect several SNPs to overlap (influence both childhood and adult adiposity). While the authors cite previous validation studies in cohorts with measured BMI in children, I noted for instance that in the HUNT study cited (Brandkvist, ref 20), the correlation coefficient for the child and adult adiposity instruments was 0.34 and the both instruments seemed to predict overweight or BMI equally well at age 16-17 years. Furthermore, in the cited Young Finns

Study (ref 19), the adult body size genetic instrument used in that study was better at predicting childhood body size (ROC AUC 0.62) than it was at predicting body size at age 34-49 (ROC AUC 0.61)! I may be wrong, but in my mind such mixture effects may bias mediation analyses. Please explain if this is not so, or that the instruments used in the current study have different properties. Please providing information on i. how many SNPs were removed in the LD clumping step to achieve independence, and ii. what was the R-squared for the each of the two instrument strengths after LD clumping?

b. Please mention and discuss the plausibility and impact of deviation from the assumptions of i. linearity in MR analyses and ii. assumption of no interactions (exposure-mediator and mediator-outcome) if relevant.

4. Magnitude of effects and comparison with other studies:

The weakness in the UKB study of coarse grouping and long recall for self report of childhood adiposity has been mentioned by the authors with references to previous validation studies and handled by some extent by harmonizing other variables by grouping (coarsening) them in 3 groups with similar proportions. However, for readers to make sense of the magnitudes of associations, the authors should specify the proportions and ideally providing information (from available sources) on the proximate mean or median BMI (or other actual measures of adiposity) in each of the three coarse categories analysed in UKB. Effect estimates in the current study are odds ratio (OR) per change in category, but these categories are not at all standard or well known. The OR in the current study cannot therefore be compared to that in the Censin study other than with respect to direction and statistical significance. (my quick and dirty estimation suggested that the average in each category is likely to differ by several units of BMI standard deviations in childhood, which suggest that the apparently large effect size is in fact for a contrast in BMI that is very large (perhaps larger than what is realistically possible to obtain in a future intervention on childhood adiposity attempting to reduce the risk of T1D, see a separate point on this below). Consider, at least in the discussion to rescale so that results are expressed in units comparable to Censin et al. 2017 (ref 15).

5. Results on other immune-associated diseases

a. Consider to mention and discuss some more results shown in table S14 that I could not see commented on in the text (or else, explain why you think this not important):

-For RA: reverse direction of child and adult adiposity instruments (significant for child and nearly so (p 0.09) for adult in the MVMR, albeit it is not clear whether these p.values were adjusted for multiple comparison.

-Ulcerative colitis (UC): inverse association in univariate analyses (albeit n.s. in MVMR).

b. Inflammatory bowel disease (IBD) is combination of UC and Crohn's Disease, but these are confusingly presented separately in tables/figures (not even side by side). I found in Suppl Table 1 that they are all from the same study, which makes sense. Please clarify this in the main tables/figures and place the three side by side for easier interpretation. The authors refer varyingly to 6, 7 and 8 other diseases, which is confusing.

MINOR COMMENTS

*Interpretation of results and implications for populations and individuals: Childhood adiposity as an explanation of the rising incidence of T1D. Norris et al. Lancet Diabetes & Endocrinology 2020 used data simulation to indicate combinations of strength of association and magnitude of change over time in an exposure that influence T1D incidence would have to be to explain a sizeable proportion of the changing incidence of T1D. The authors should put the size of their estimate in the context of a plausible estimate of the change in adiposity in the population over the past decades or so, and also discuss the plausibility for an individual to change his/her body size to such an extent that it would lead to a meaningful change in risk of T1D. My hunch is that the current results suggest limited practical implications, although the academic interest in understanding causal risk factors for T1D is

still significant.

*Forest plots were informative, but open and closed circles was used inconsistently and not explained.

*the comment on Sardinia study as the only one where results differed. This was also the one with lowest precision/wides confidence intervals. Was there any evidence of interaction between exposure and study/site?

*suppl fig 2 and 3. Study stratified effect estimates. Since the authors report odds ratios as effect estimates in the main manuscript, I would have expected this also for suppl fig 2 and 3, but suspect effect estimates in these figs are something else – perhaps log OR? Please specify and harmonize if possible (possibly by showing ORs but adjusting scale to be log). Also Supplemental tables report Coeff, but ORs in the table. Please report also ORs in suppl Tables where relevant

*index of suppl tables say Table 13 and 14 include information about 8 types of autoimmune diseases. First, many of the diseases are not considered autoimmune (eg asthma, eczema). Second, the tables does not show 8 diseases, it shows 6 diseases (as IBD, I believe) IBD is pool of UC and CD, and T2D is not shown. Please correct

*line 313: 10 of 16 analyses in Table S13 significant. This should be 10 of 14, or else T2D results should be added to the table.

* Suppl tables S13 and S14 are a bit confusing but appears to show results for different age at onset of outcomes? Two lines per disease, where the column with the heading outcome contains "age_10" or adult". I suspect that the age_10 vs adult refers not to the outcome but rather the exposure instrument? The heading of the column specifying diseases is headed by "exposure" but this should probably be "outcome". Also: the univariate MR results in Table S13 seems also to be contained in Table S14 - why have a separate S13? (except the column with heading FDR , which appear to be the q-value of the univariate MR (?) – this could also be included in S14 and S13 removed.

*In the first paragraph of the discussion, the authors say their results support previous observational studies, but no references are cited. Consider citing at least a few well designed studies supporting this claim.

*the authors used the terms (child or adult) adiposity and body size in the manuscript, but it is not always clear whether it refers to the same concept or not.

*Please provide reference to BOLT-LMM software, as this is not likely generally known.

Reviewer #3:

Remarks to the Author:

This Mendelian analysis seems to confirm the predictive association of childhood obesity with T1D and suggests it is causative in GWAS cohorts ,despite the crude measures of obesity and height These results are consistent with papers in the literature from a variety of T1D prospectively followed cohorts which are not referenced here A strength is the inclusion of other autoimmune diseases for comparison. However a major weakness is not using the age at T1D diagnosis as the outcome and I think this publication should await the availability of those data As it stands it appears that adult weight could be included in the prediction of childhood diabetes and lack of association of adult obesity would be expected

Other points

- 1 It is frustrating to have to read other references in order to get a basic understanding of the cohorts examined
- 2 in order to assess the results, please provide the prevalence of T1D in the various cohorts and overall prevalence
3. Please review the current literature on the topic and include those references including some on the relationship of BMI with IA progression and correct the statement that there are no studies
- 4 There are a number of statements in the manuscript that need references and also some need to be checked to be sure they are appropriate
- 5 it appears that reference to figs 1 a,b and c may refer to ref 11. This is confusing
- 6 There is some literature suggesting that birthweight is related to HLA. Are you suggesting you did not find this?.
- 7

REVIEWER COMMENTS

Reviewer #1 (Remarks to the Author):

Richardson et al present a Mendelian Randomisation analysis to estimate the relative causal effect sizes of childhood and adult BMI on Type 1 Diabetes as well as a number of other autoimmune diseases. This is a well written and interesting study addressing an important question, finding a causal role for childhood BMI on T1D, while the role of adult BMI was more important for the majority of the other diseases tested. I have some minor comments listed below:

1. The authors used UK Biobank to select genetic variants to use as instrumental variables for adult BMI by categorising the raw continuous BMI variable to be consistent with the childhood BMI instruments. The authors also used UK Biobank to select instruments for birth weight, but I was unclear as to whether the birthweight phenotype was categorised in the same way as the adult BMI phenotype. The methods suggest that it was not. Could the authors clarify whether the birthweight phenotype was categorised or not, and if not why was the decision taken to categorise adult BMI but not birthweight?

Thank you for your suggestions to help refine our manuscript. Birthweight was not categorised in our analyses due to only n=280,232 participants reporting this in UKB. This would therefore mean that categorizing them in the same manner as adult BMI (available in nearly all study participants) would lead to a reduction in sample size. This is because our harmonization approach involved making measures comparable to the childhood body size variable which requires both measures for all participants analysed. We therefore left birthweight continuous to retain instrument derivation for childhood/adult body in the maximum sample size available. This has now been clarified on page 8:

'The same analysis pipeline was applied to generate genetic instruments for birthweight which was kept continuous due to only being available on a total of 261,932 UKB individuals. This trait was rank-based inverse normal transformed to ensure normality and adjusted as before for age, sex and genotyping chip.'

2. Results line 261: The authors write "Beta=-0.007 per standard deviation change in BMI" which suggests that this result is from an MR of the effect of BMI on T1D but the text suggests that this result is from an MR of the effect of T1D on BMI – should the units not be SD change in BMI per log odds T1D liability?

Thank you for spotting this typo – we have repeated our T1D GRS analysis and updated this section accordingly (page 13):

'We also identified limited evidence of a reverse direction of effect between T1D genetic liability and childhood body size (Beta=0.002 per 1-SD change in T1D liability, 95% CI=-0.001 to 0.005, P=0.236), meaning that the effect of childhood body size on T1D is unlikely to be explained by reverse causality. There was also little evidence to suggest that T1D genetic

liability has an effect on BMI in adulthood (Beta=-0.002, 95% CI=-0.005 to 0.001, P=0.266) (Supplementary Table 7).'

3. In the supplemental tables it was not immediately clear to be whether effect size estimates given were odds ratios or log-odds ratios – these were often just labelled “Beta” or “estimate”. Could the authors clarify in table captions what units the figures given relate to?

We have now updated the captions in supplementary tables to clarify this.

4. Line 313: “Using univariable MR, 10 of the 16 analyses undertaken provided evidence that adiposity in either childhood or adulthood influenced chronic immune disease risk based on FDR<5% (Supplementary Table 13)” – Table S13 seems to only include 14 analyses (7 traits, each for childhood and adulthood BMI)

Thank you again for identifying this typo which we have now corrected on page 18:

‘Using univariable MR, 9 of the 14 analyses undertaken provided evidence that adiposity in either childhood or adulthood influenced chronic immune disease risk based on FDR<5% (Supplementary Table 16).'

5. Could the authors comment on how confident they are that the estimates of childhood and adult BMI effects on for example Asthma. While the effect estimate for childhood BMI includes the null but the adult BMI estimate does not, the effect estimates are similar, and their confidence intervals include each other. Does this affect the interpretation that childhood BMI only has indirect effects on asthma?

In line with comments from other reviews we have now incorporated FDR corrections into our multivariable analyses. Both direct and indirect effect estimates on asthma risk are not robust to FDR<5% corrections and we have therefore changed the interpretation of these results to address this comment. This is now discussed on page 18:

‘There was stronger evidence however that childhood body size indirectly influences disease risk via adult body size on: asthma risk (OR=1.30, 95% CI=1.04 to 1.63, P=0.022), dermatitis and eczema (OR=1.30, 95% CI=1.03 to 1.64, P=0.026) and hypothyroidism (OR=1.94, 95% CI=1.45 to 2.61, P=9.64x10⁻⁶). After correcting multivariable analyses for false discovery rate (FDR) only the effect on hypothyroidism remained robust (FDR=1.35x10⁻⁴).'

6. In the discussion, line 375 the authors state that the fact that the MR-Egger 95% confidence interval for the adult BMI and birthweight effects on T1D overlap the null suggest evidence of “horizontal pleiotropy”. It is my understanding that the intercept of the MR-Egger analysis is a measure of the presence of unbalanced horizontal pleiotropy, and that MR-Egger is known to be less powerful than IVW due to the additional degree of freedom in the estimation of both the slope and intercept parameters. Is the interpretation that the MR-Egger estimate includes the null indicates the presence of pleiotropy correct?

We have toned this statement down in light of your comment. The reviewer is correct that the inclusion of the null in MR-Egger analyses may not necessarily indicate the presence of pleiotropy, although taken together with findings from the other approaches does suggest that estimates for adult body size and birthweight were not as strongly supportive of these risk factors having an effect on T1D risk in our study (page 21):

'However, estimates derived using the MR-Egger method only supported the childhood body size effect (OR=5.06, 95% CI=1.52 to 16.81, P=0.009), whereas confidence intervals for both birthweight and adult body size overlapped with the null meaning that they were not as strongly supported by this approach of having a genetically predicted effect on T1D risk.'

7. The authors include birthweight in their multivariable MR analysis. They state that the reason for this is to investigate whether individual's body size in very early life may be responsible for the effects identified using childhood BMI instrument. While they acknowledge that their analyses do not account for maternal effects, they give little discussion to how much impact this could have on their conclusions. For instance, the correlation between maternal and fetal genotype can lead to both under- and over-estimation of SNP-exposure and SNP-outcome associations. If these over- or under-estimations result in an under-estimation of the causal effect size of birthweight on T1D this could lead to an incorrect conclusion that birthweight does not causally impact on T1D risk but childhood BMI does. As such, I would like to at least see some consideration of the impact of this on the results in the discussion, and I also feel that the statement in the first line of the discussion "We present evidence that adiposity in childhood increases the risk of T1D independently of body size at birth and adulthood" should be toned down in relation to size at birth.

We have toned down the opening sentence of our discussion in line with your suggestion (page 20):

'We present evidence suggesting that adiposity in childhood increases the risk of T1D based on the age-at-diagnosis of the participants analysed in this study (mean age=16.57 years).'

Furthermore, we have further emphasized in our discussion that further research is required to formally estimate the causal effect of birthweight on T1D using a family design (page 21):

'We incorporated birthweight as an additional exposure in our multivariable model to assess whether it may help explain effect of childhood body size on T1D. As our estimates remained robust, these findings do not seem to suggest that variation in birthweight is responsible for the effect of genetically predicted childhood body size on T1D risk identified in our analysis. However, a more appropriate evaluation of the influence of birthweight on T1D risk requires in-depth evaluation using both maternal and fetal genetic effects, as undertaken previously, once sample sizes of both maternal and offspring T1D cases are sufficient^{36,49}. Amongst other sources of bias, future endeavours applying this study design will be able to investigate

whether our results may be underestimating the genetically predicted effect of birthweight on T1D risk.'

8. The authors mention that adult BMI is unlikely to causally impact T1D risk given the age of onset of this disease, however T1D does not occur only in childhood and can manifest only in later life. Could the authors comment in the discussion of whether their results suggest that childhood BMI a more important causal factor than adult BMI in later onset T1D, or whether the fact that the majority of cases in the T1D GWAS would have had early onset mean that these results are not generalisable to later onset T1D?

Thank you again for this useful suggestion. We have now added to the discussion that future work evaluating the effect of adult adiposity of later life T1D diagnosis could potentially provide evidence of a causal relationship (page 21):

'Further work is required to investigate late-onset T1D using age-at-diagnosis data once it becomes available in large sample sizes, particularly given the challenges of T1D diagnosis in adulthood⁴⁸. This would be particularly valuable in investigating whether adiposity in adulthood could increase risk of late-onset T1D, which our study may be underpowered to detect due to the large majority of individuals in our T1D sample being diagnosed during childhood.'

Reviewer #2 (Remarks to the Author):

The study uses several very large data sets to conduct state of the art mendelian randomization (MR) analyses attempting to disentangle the potential direct and indirect causal effects of childhood and adult adiposity on type 1 diabetes T1D and other immune-mediated diseases such as asthma, inflammatory bowel diseases (and extending previous results on type 2 diabetes). The main conclusions are that adiposity in childhood rather than in adulthood is directly causally related to the risk of type 1 diabetes, while the opposite seems to be the case for T2D, asthma, eczema and hypothyroidism. The T1D result confirms and extends a previous finding by Censin using univariate MR analyses (ref 15 in the manuscript).

The authors are top experts and analyses seems largely well done, and the paper contributes some novel results. My most important concern is that I struggle with the idea that adult adiposity should mediate the effect of childhood adiposity, or indeed have any causal effect on diseases with typical onset in childhood, be it T1D, asthma or eczema. The authors could do a better job at explaining to readers how this makes sense. I have some additional comments as well as a number of mostly minor corrections or suggestions. (some of the authors have recently contributed several other interesting papers using similar approaches for type 2 diabetes and other disease outcomes typically occurring in adulthood, where it makes more sense to me).

COMMENTS

1.Role of adult BMI in early onset disease: First let me say that although I have a decent

understanding of traditional mediation analyses and simple MR analyses, I have no first hand analysis experience with multivariable MR (MVMR) for mediation analysis. My comments may reflect this, but I believe this is also likely to be the case for 99% of readers of this paper.

a. On line 368-370, the authors write: “As expected given the average age-at-diagnosis of T1D, the effect of childhood body size remained robust after accounting for adult body size using a much larger number of genetic instruments than previously used (n=280 in this study versus n=13 previously)”. In a way, this states that the main research question is not interesting, as it is obvious that adult BMI cannot influence a disease which typically starts and is diagnosed in childhood. I believe this argument can be put forward also for asthma, eczema and perhaps some of the other immune-mediated diseases outcomes studied. Please explain. Is there something with the MR study that should make us think differently about how to interpret such mediation?

Thank you for your comprehensive review of our manuscript to help refine it. A major strength of our study is the application of human genetics to disentangle the effects of body size at two separate stages in the lifecourse on T1D risk. This approach is a recent development in the MR field which we have pioneered in the last couple of years and its application to T1D risk in this study is further validation of its utility. As the reviewer suggests, we would not necessarily expect adult body size to influence T1D risk the average age at diagnosis of the participants involved in this work. However, our application of multivariable MR to robustly demonstrate this is an important showcase that our model is capable of separating the effects of childhood body size from adult body size. This improves the robustness of our conclusions that childhood body size is a causal risk factor for T1D risk, which has important implications. Moreover, evidence from this method benefits from the other strengths of MR, such as estimates being less prone to confounding and reverse causation than observational studies. In fact, estimating the effect of childhood body size on T1D risk in an observational study would be extremely challenging to investigate (particularly given its prevalence).

We have therefore added some discussion around this point to page 22:

‘Firstly, the use of genetic variation in a two-sample MR framework allowed us to analyse a large number of genetic instruments from the UK Biobank sample for body size (n=454,023) with a meta-analysed sample of T1D cases (up to n=15,573), almost twice the number of cases used in a previous study¹⁷. As such our results are less prone to bias attributed to reverse causation and confounding factors compared to more traditional epidemiology approaches. Furthermore, this study design allowed us to investigate the direct and indirect effects of childhood body size on T1D as well as seven other chronic immune-associated diseases in turn, which would be extremely challenging to undertake without the use of human genetics.’

b. Information on age at diagnosis: I believe most readers would appreciate a table of characteristics which at least contains the mean and range of age at diagnosis. Some details

on diagnostic criteria should be provided, e.g. if T1D in adults how was diabetes classified. Any diagnoses such as asthma based on self- or parent report?

We have now added information on age at diagnosis for T1D cases to pages 8 and 9 of the manuscript, and a new Supplementary Table 1 with cohort breakdown by age:

'We firstly applied our multivariable approach using a large number of childhood and adult body size instruments to T1D data analysed previously in the study by Censin et al. (n=5,913 cases diagnosed before the age of 17 years and n=8,828 controls). Results from this analysis were then validated using a recent large-scale meta-analysis of up to 15,573 cases and 158,408 controls²⁵. Analyses were then repeated separately in each contributing cohort from this meta-analysis: Illumina genotyped UK samples (3,983 cases and 3,994 controls), Affymetrix genotyped UK samples (1,926 cases and 3,342 controls), Sardinians (1,558 cases and 2,882 controls), Finnish FinnGen samples (4,933 cases and 148,190 controls) and the T1DGC European-ancestry family sample (3,173 affected-offspring trios, analysed by the transmission disequilibrium test).'

In terms of age-at-diagnosis, 7,453 T1D meta-analysis cases were diagnosed before 10 years of age, 4,368 between 10 and 20 years old, 3,352 over 20 years old and 400 with missing data) (see Supplementary Table 1 for a breakdown by cohort). As such nearly half of cases included in the meta-analysis had age-at-diagnosis later than 10, the age at which our childhood body size instrument is based on.'

We have also conducted further analyses to demonstrate that, although the childhood recall data is based on 'age 10' in the lifecourse, that the childhood score relates preferentially to body mass index across multiple timepoints in childhood in comparison to the adult body size score.

Pages 11:

Additionally, we derived a genetic risk score (GRS) using data from the ALSPAC cohort and investigated the effect of T1D genetic liability on measured BMI data from the mean age 9.9 years (range=8.9 to 11.5 years old) clinic. We also used data from the ALSPAC cohort to evaluate how our childhood and adult body size instruments relate to BMI at 12 timepoints in childhood prior to age 10.

And page 13:

'Investigating how our childhood and adult body size instruments relate to measured BMI in the ALSPAC cohort found that the childhood body size score associates more strongly with BMI at mean age 9.9 years but also at 11 other earlier timepoints in the lifecourse (Supplementary Figure 2).'

The following has also been added to the Discussion on page 23 to highlight the age at diagnosis in our meta-analysis:

'A further limitation of our study is that childhood body size was measured at age 10, while 47.9% of our T1D meta-analysis cases were diagnosed before 10. Among this subset of cases, we cannot eliminate the possibility that exposure to obesity occurred after developing T1D, which would preclude a causal relationship. However, we believe our study to have good statistical power due to a) 49.5% of cases having known age-at-diagnosis older than 10 and b) obesity at age 10 being presumably correlated with obesity at earlier ages, providing effective exposure prior to disease diagnosis for a larger subset of T1D cases. FinnGen effect estimates were similar to most other cohorts despite containing a far greater proportion of cases diagnosed over age 20 (62%), supporting (b).'

c. Analyses by age at diagnosis: Under statistical analysis (line 200), the authors wrote: "We included adult body size to demonstrate the importance of using genetic scores to separate the effects of adiposity at different stages in the life course when investigating either early or late onset disease outcomes". I found no analysis of early vs late onset of a single disease, nor any explanation which of the specific diseases were considered early or late onset. At least for type 1 diabetes and the most important contributor to the genetic liability, HLA, there is evidence that the strength of association varies by age at onset. Analyses by age at diagnosis of T1D and perhaps other diseases would probably be informative.

Thank you for this suggestion. We have firstly added information on age at diagnosis for T1D as described above, but have also now added information for the other disease outcomes investigated in this study to Supplementary Table 2. However, analyses stratified by age at diagnosis of T1D would be extremely difficult to undertake for this study. It would involve a substantial amount of time and work to undertake a T1D genome-wide association study stratified by age at diagnosis data and involve coordinating an effort with many international research groups. Even though this may ultimately allow us to demonstrate that adult adiposity is a risk factor for late onset T1D, it will not diminish the evidence identified in our current study where the aim was to investigate the causal influence of childhood adiposity of T1D risk. We have added this discussion to page 22:

'Firstly, the use of genetic variation in a two-sample MR framework allowed us to analyse a large number of genetic instruments from the UK Biobank sample for body size (n=454,023) with a meta-analysed sample of T1D cases (up to n=15,573), almost twice the number of cases used in a previous study¹⁵. As such our results are less prone to bias attributed to reverse causation and confounding factors compared to more traditional epidemiology approaches. Furthermore, this study design allowed us to investigate the direct and indirect effects of childhood body size on T1D as well as seven other chronic immune-associated diseases in turn, which would be extremely challenging to undertake without the use of human genetics.'

d. Given the above, I think the analysis accounting simultaneously for birth weight and childhood adiposity makes more sense. I appreciate the authors comment that BWT was not appropriately handled to include maternal effects but I think results similar to those done for T1D should be done also for the other disease outcomes (including T2D). My hunch is that although there may be maternal effects influencing birth weight, the fetal genes (inherited maternal and paternal) are likely most important in such a mediation

analysis. Also, interpretation of the apparent inverse association of birth weight with T1D is difficult given previous observational studies suggesting weak but very consistent positive association across many large studies.

We have discussed on page 21 that evidence of an inverse effect between genetically predicted birthweight and T1D risk was not supported by the MR-Egger, which limits the strength of evidence provided by the IVW method:

'However, estimates derived using the MR-Egger method only supported the childhood body size effect (OR=5.06, 95% CI=1.52 to 16.81, P=0.009), whereas confidence intervals for both birthweight and adult body size overlapped with the null meaning that they were not as strongly supported by this approach of having a genetically predicted effect on T1D risk.'

We are also grateful that the reviewer appreciates our comments regarding the necessity of a family design to appropriately investigate the effect of birthweight on disease risk once these data are available. However, given that T1D was the only outcome analysed where we identified evidence of a direct effect for childhood body size, we are unsure of the rationale regarding the analysis of birthweight of all other endpoints in this study. This is discussed on page 11:

'Birthweight was analysed in this study to investigate whether an individual's body size in very early life (e.g. before age 5 years) may be responsible for the effects identified using our childhood genetic score (Supplementary Figure 1).'

As such, we are not sure why accounting for birthweight is a necessary addition if there is weak evidence that childhood body size directly influences disease. We do agree that the role of birthweight in adult disease risk is an important but separate research question to the one investigated in our study which we have discussed on page 21:

'We incorporated birthweight as an additional exposure in our multivariable model to assess whether it may help explain effect of childhood body size on T1D. As our estimates remained robust, these findings do not seem to suggest that variation in birthweight is responsible for the effect of genetically predicted childhood body size on T1D risk identified in our analysis. However, a more appropriate evaluation of the influence of birthweight on T1D risk requires in-depth evaluation using both maternal and fetal genetic effects, as undertaken previously, once sample sizes of both maternal and offspring T1D cases are sufficient^{36,49}. Amongst other sources of bias, future endeavours applying this study design will be able to investigate whether our results may be underestimating the genetically predicted effect of birthweight on T1D risk.'

2. Reverse causation:

a. Effect of T1D genetic liability. Please provide more information on the GRS (in addition to citing a reference). How strongly does it predict T1D (eg. AUC), how were HLA SNPs handled – haplotypes, interactions etc?

A list of all genetic variants included in the T1D GRS have now been included in Supplementary Table 3. We have now updated the instruments included in our T1D GRS based on a recent study (Crouch et al) where the authors identified 63 genetic variants with $P < 5 \times 10^{-8}$. As this is an MR study, instead of conducting AUC to investigate instrument strength we have calculated F-statistics ($F=196$) which suggests our analyses are unlikely to be prone to weak instrument bias. Furthermore, as suggested we have described how the 5 HLA SNPs in this score have been handled (page 11):

'In this analysis we used a set of 63 genetic instruments for T1D identified from a recent meta-analysis (of up to 15,573 cases and 158,408 controls²⁵) (Supplementary Table 3) which had an F-statistic of 196. Non-HLA SNPs in this score were selected to be independent (pairwise $r^2 < 0.01$) via an LD-clumping procedure, while the 5 HLA SNPs had pairwise $r^2 < 0.02$ in European non-Finnish 1000 Genomes samples (using the LDlink online tool, <https://ldlink.nci.nih.gov/>).'

b. On line 382, the authors write: "Whilst we did not find evidence that genetic liability towards T1D may influence childhood adiposity, your results suggest that it may have an effect on lower body size in adulthood based on the MR-Egger and weighted median methods. Medical practitioners promote healthy living among T1D patients in order to keep HbA1c levels low, which is one possible explanation for this result." This is relevant for the general interpretation of MR results and mediation. My take on the author's interpretation is that they believe the causal effect to be mediated via diagnosis of T1D and treatment of the disease (besides the fact that intensive insulin treatment to lower HbA1c will in fact tend to increase BMI, not lower it). I am wondering if this is a plausible interpretation at all, because in the general population, the incidence and prevalence of T1D is low (<1%), and if most or all of the effect of a T1D genetic risk score (GRS) was really mediated via diagnosis and treatment of T1D, I believe that this would not be detectable as a "total effect" in the population, given the low prevalence of the mediator. Unless I have misunderstood how these MR analyses should be interpreted, please rephrase this part of the discussion, and provide a more plausible explanation of this finding.

After updating our T1D GRS as described above in line with other comments, we found that there is no longer strong evidence of an effect of T1D liability on adult-body size and have therefore removed this section from the discussion.

c. Estimating potential causal effect of T1D GRS on BMI in childhood: These analyses could preferably have been done in cohorts with measured childhood BMI (HUNT, ALSPAC, YFS, studies some of the authors have used previously for similar analyses of other outcomes), rather than the UKB where childhood adiposity around the age of 10 years was recalled (from mean age around 56 years at study enrolment) and categorised in three general categories, which is a weakness (both potential recall bias and coarse categories).

Thank you for this excellent suggestion. We have now included a further MR analysis of T1D genetic liability on measured childhood BMI data from the ALSPAC cohort at

mean age=9.9 years as the reviewer has suggested. This has now been reported on page 11:

'Additionally, we derived a genetic risk score (GRS) using data from the ALSPAC cohort and investigated the effect of T1D genetic liability on measured BMI data from the mean age 9.9 years (range=8.9 to 11.5 years old) clinic.'

And page 13:

'Conducting this MR analysis using data from the ALSPAC cohort supported these findings using measured childhood BMI at mean age 9.9 years in the lifecourse (Beta=0.033 per 1-SD change in T1D GRS, 95% CI=-0.040 to 0.106, P=0.382).'

d.Results for “converse” effect: T1D GRS on adult BMI was reported as -0.007 per change in SD of BMI. This must be an error, as the analysis is change in BMI (presumably in units of SD, or what?) per change in (units?) of T1D GRS! Please correct and clarify.

Thank you for spotting this typo. We have updated this sentence in line with new analysis recommended by reviewers. This is now described on page 13:

'We also identified limited evidence of a reverse direction of effect between T1D genetic liability and childhood body size (Beta=0.002 per 1-SD change in T1D liability, 95% CI=-0.001 to 0.005, P=0.236), meaning that the effect of childhood body size on T1D is unlikely to be explained by reverse causality. There was also little evidence to suggest that T1D genetic liability has an effect on BMI in adulthood (Beta=-0.002, 95% CI=-0.005 to 0.001, P=0.266) (Supplementary Table 7).'

e.Please also discuss potential scenarios outlined by Burgess and Swanson (Int J Epidemiol 2020) where MR analyses can be influence by reverse causation type of biases - relevance for the current results?

Apologies as we were unable to identify the study in the IJE referred to by the reviewer. We did however find this study by the same authors earlier this year in the EJE (but please correct us if this is incorrect):

<https://link.springer.com/article/10.1007/s10654-021-00726-8>

We agree with this article that MR analyses, whilst robust to reverse causation, are not entirely immune from it. This is described in the original version of this work on page 22:

'As such our results are less prone to bias attributed to reverse causation and confounding factors compared to more traditional epidemiology approaches.'

Additionally on page 23 we have discussed potential scenarios outlined by this article are unlikely to have introduced bias into the findings of our work:

'Additionally,, although MR studies are typically considered to be less prone to reverse causation than observational studies, there are possible scenarios where this could still bias findings as outlined in a recent review⁵⁶. This is why in this study we investigated the converse direction of effect for our primary analysis using MR i.e. whether T1D genetic liability influences childhood body size. As weak evidence of an effect was found in this sensitivity analyses, our findings suggest that T1D resides downstream of childhood BMI and also that a scenario involving feedback mechanisms are unlikely. Accounting for birthweight in our model also mitigates the likelihood that a cross-generational effect is underlying the genetically predicted effect of childhood body size on T1D risk found in our study.'

3. Assumptions and interpretation of MVMR mediation analyses :

a. uncorrelated instruments: In traditional mediation analysis, the exposure and mediator can be correlated, whereas MVMR mediation analysis apparently requires uncorrelated instruments for childhood and adult adiposity. I would expect several SNPs to overlap (influence both childhood and adult adiposity). While the authors cite previous validation studies in cohorts with measured BMI in children, I noted for instance that in the HUNT study cited (Brandkvist, ref 20), the correlation coefficient for the child and adult adiposity instruments was 0.34 and the both instruments seemed to predict overweight or BMI equally well at age 16-17 years. Furthermore, in the cited Young Finns Study (ref 19), the adult body size genetic instrument used in that study was better at predicting childhood body size (ROC AUC 0.62) than it was at predicting body size at age 34-49 (ROC AUC 0.61)! I may be wrong, but in my mind such mixture effects may bias mediation analyses. Please explain if this is not so, or that the instruments used in the current study have different properties. Please providing information on i. how many SNPs were removed in the LD clumping step to achieve independence, and ii. what was the R-squared for the each of the two instrument strengths after LD clumping?

Along with the various forms of validation analyses mentioned by the reviewer, we also conducted LD score regression analyses which further support the separation between our childhood and adult genetic scores (page 7):

'Other validation analyses have also been conducted previously, whereby GWAS results for the childhood measure had a higher genetic correlation with measured childhood obesity from an independent sample ($r_G=0.85$) compared to the adult measure ($r_G=0.67$). Conversely, genome-wide estimates for the adult measure were more strongly correlated with measured BMI in adulthood ($r_G=0.96$) compared to the childhood measure ($r_G=0.64$)¹³.'

However, the most appropriate approach to demonstrate the capability of our genetic scores to separately predict body size at two different timepoints in the life course come in the form of conditional F-statistics as outlined in the MVMR study we have referenced by Sanderson et al (reference 38 in our manuscript). We calculated these in the original study for our instruments which supported their robustness against weak instrument bias. This is additionally discussed on page 7:

'Furthermore, using these instruments previously for multivariable MR provided F-statistics > 10 suggesting that derived results are unlikely to be prone to weak instrument bias¹³.'

These are preferred over R-squared values for instrument strength after clumping our scores. We have also now added some text to clarify to readers how many SNPs were dropped during LD clumping to ensure independence of instruments (page 9):

'In total, there were 280 childhood body size and 515 adult body size instruments available for analysis after harmonization with T1D genetic estimates, where 81 were subsequently removed prior to conducting multivariable MR analyses.'

b. Please mention and discuss the plausibility and impact of deviation from the assumptions of i. linearity in MR analyses and ii. assumption of no interactions (exposure-mediator and mediator- outcome) if relevant.

We investigated the assumption regarding linearity across exposures in the original study where our instruments were derived and did not identify strong evidence against this assumption. This is now discussed on page 7:

'In the original study where these instruments were derived we did not identify any evidence against a linear relationship between our exposure variable in line with the assumptions of multivariable MR¹³.'

As our results suggest that childhood body size has a direct effect on T1D risk (i.e. not indirectly via a mediator) we do not believe that the assumption of no interaction is possible to investigate.

4. Magnitude of effects and comparison with other studies:

The weakness in the UKB study of coarse grouping and long recall for self report of childhood adiposity has been mentioned by the authors with references to previous validation studies and handled by some extent by harmonizing other variables by grouping (coarsening) them in 3 groups with similar proportions. However, for readers to make sense of the magnitudes of associations, the authors should specify the proportions and ideally providing information (from available sources) on the proximate mean or median BMI (or other actual measures of adiposity) in each of the three coarse categories analysed in UKB. Effect estimates in the current study are odds ratio (OR) per change in category, but these categories are not at all standard or well known. The OR in the current study cannot therefore be compared to that in the Censin study other than with respect to direction and statistical significance. (my quick and dirty estimation suggested that the average in each category is likely to differ by several units of BMI standard deviations in childhood, which suggest that the apparently large effect size is in fact for a contrast in BMI that is very large (perhaps larger than what is realistically possible to obtain in a future intervention on childhood adiposity attempting to reduce the risk of T1D, see a separate point on this below). Consider, at least in the discussion to rescale so that results are expressed in units comparable to Censin et al. 2017 (ref 15).

We have estimated the genetically predicted change in BMI per change in body size category using results from the early growth genetics (EGG) GWAS that the Censin et al study used to identify MR instruments for. Our results provide another compelling proof of concept for our childhood and adult body size instruments:

Exposure	Outcome	Univariable MR			Multivariable MR	
		Beta	SE	P	Beta_MV	SE_MV
Childhood body size	Childhood BMI (EGG)	1.677	0.071	3.86E-122	1.490	0.10
Adult body size	Childhood BMI (EGG)	1.110	0.092	1.02E-33	0.360	0.09

In terms of comparing effect sizes of estimates, this means that the OR=2.05 identified in our analysis on T1D risk equates to an OR=1.23 per 1-SD change using the EGG BMI estimates. This is lower in magnitude compared to the Censin et al study (OR=1.32), although an important consideration here is that the UKB participants were born between 1934 and 1971 (mean=1952), where the average childhood BMI would have been drastically different to the mean year of birth for cohorts contributing to the EGG GWAS (most involved participants born in the 1990s). This makes direct comparisons between our estimates with Censin et al's challenging.

5. Results on other immune-associated diseases

a. Consider to mention and discuss some more results shown in table S14 that I could not see commented on in the text (or else, explain why you think this not important):

-For RA: reverse direction of child and adult adiposity instruments (significant for child and nearly so (p 0.09) for adult in the MVMR, albeit it is not clear whether these p.values were adjusted for multiple comparison.

-Ulcerative colitis (UC): inverse association in univariate analyses (albeit n.s. in MVMR).

We have now added FDR corrections to this table and although the univariable analyses are FDR<5% for RA the direct and indirect estimates of childhood body size are not robust to these corrections in the MVMR analysis. Whilst exploring the relationship between body size and RA may be of interest to future research, we feel that this is outside the scope of this manuscript given our focus on T1D. The other immune-associated diseases and disorders were analysed in this study to explore whether the direct effect of childhood body size on T1D was generalizable to other endpoints related to the immune system as described on page 4:

'Lastly, it has not yet been investigated whether the effect of childhood body size on T1D risk represents a more generalizable effect on the immune system which may additionally impact other types of immune-associated or autoinflammatory diseases. If there is a T1D-specific effect, this would suggest early life β -cell fragility stemming from diet-induced metabolic stress is likely to be a causal pathway through which childhood body size leads to increased T1D risk.'

b. Inflammatory bowel disease (IBD) is combination of UC and Crohn's Disease, but these are confusingly presented separately in tables/figures (not even side by side). I found in

Suppl Table 1 that they are all from the same study, which makes sense. Please clarify this in the main tables/figures and place the three side by side for easier interpretation. The authors refer varying to 6, 7 and 8 other diseases, which is confusing.

We have now moved IBD, UC and CD so that they are side by side in both table and figures. We have clarified that 7 immune-associated endpoints were analysed along with T1D. The following has been added to page 9:

'We also obtained estimates using results from a GWAS of T2D, updated since our previous study²⁶, and seven of the most common immune-associated disease endpoints: asthma, atopic dermatitis and eczema, hypothyroidism, rheumatoid arthritis, inflammatory bowel disease and its two subtypes (Crohn's disease and ulcerative colitis).'

MINOR COMMENTS

*Interpretation of results and implications for populations and individuals: Childhood adiposity as an explanation of the rising incidence of T1D. Norris et al. Lancet Diabetes & Endocrinology 2020 used data simulation to indicate combinations of strength of association and magnitude of change over time in an exposure that influence T1D incidence would have to be to explain a sizeable proportion of the changing incidence of T1D. The authors should put the size of their estimate in the context of a plausible estimate of the change in adiposity in the population over the past decades or so, and also discuss the plausibility for an individual to change his/her body size to such an extent that it would lead to a meaningful change in risk of T1D. My hunch is that the current results suggest limited practical implications, although the academic interest in understanding causal risk factors for T1D is still significant.

We have created a comprehensive table (Supplementary Table 14) giving the proportions of plumper than average, average and thinner than average individuals, broken down into T1D affected and T1D unaffected individuals, as a function of several population parameters, and using a relative risk (RR) of 2.64, taken from the OR from our MR-Egger meta-analysis result (assuming that the OR and RR are approximately equal). The table shows that by changing the frequency of plumper than average individuals from 0.15 to 0.05, and keeping the other categories equal, the prevalence of T1D drops by a factor of 1.22, which is quite significant, and could go somewhat to explaining the recent increases in rates of T1D diagnosis.

In the main text we have added the following to page 14:

'Assuming that our OR estimates are approximately equal to Relative Risks (RRs), and assuming a T1D prevalence of 0.5%, we used our direct childhood body size MR-Egger OR estimate from the meta-analysis (OR=2.64) to build a table of proportions for T1D affected and unaffected individuals lying in each body size category. Mimicking an intervention, we changed the proportion of individuals who are 'plumper than average' from 0.159, the proportion within our data, to 0.059, producing a fall in T1D prevalence to 0.39% (a 22% reduction) (Supplementary Table 14).'

We have also mentioned this in the discussion on page 20:

'Regardless of the underlying mechanisms, our findings suggest that a critical window exists in childhood to mitigate the influence of adiposity on the escalating numbers of T1D diagnoses, and that an approximately 22% reduction in the number of T1D cases is plausible if the proportion of children within the highest obesity category were to be reduced by 10%, from 15.9% to 5.9%.'

*Forest plots were informative, but open and closed circles was used inconsistently and not explained.

We have now added the following sentence to the legends in Figures 2, 3 and 4 to clarify this:

'Central estimates are illustrated as circles which were filled when confidence intervals did not overlap with the null.'

*the comment on Sardinia study as the only one where results differed. This was also the one with lowest precision/wides confidence intervals. Was there any evidence of interaction between exposure and study/site?

While it is not possible to investigate this further using the data available, it is well known that rates of obesity in Sardinia are lower than in the countries where the other cohorts were collected, and this is probably due to lower exposure to obesogenic diets. As childhood obesity instruments are likely to interact with dietary exposures, this is a plausible explanation for why the Sardinian cohort demonstrated weaker MR associations between childhood obesity and T1D. This is also the smallest cohort in our meta-analysis, which also likely contributes to the wider confidence interval for the corresponding estimate. However, we see this as too speculative to include in the revised text. The direction of effect only diverged from the other cohorts when using the weighted median method, and taking a consensus of all three MR methods we would conclude that there is no reason to believe that the Sardinian effect is much lower than the other cohorts' effects.

We have included the following on the sizes of the cohorts (page 8) (as well as adding this information to Supplementary Table 1):

'Analyses were then repeated separately in each contributing cohort from this meta-analysis: Illumina genotyped UK samples (3,983 cases and 3,994 controls), Affymetrix genotyped UK samples (1,926 cases and 3,342 controls), Sardinians (1,558 cases and 2,882 controls), Finnish FinnGen samples (4,933 cases and 148,190 controls) and the T1DGC European-ancestry family sample (3,173 affected-offspring trios, analysed by the transmission disequilibrium test).'

*suppl fig 2 and 3. Study stratified effect estimates. Since the authors report odds ratios as effect estimates in the main manuscript, I would have expected this also for suppl fig 2 and

3, but suspect effect estimates in these figs are something else – perhaps log OR? Please specify and harmonize if possible (possibly by showing ORs but adjusting scale to be log). Also Supplemental tables report Coeff, but ORs in the table. Please report also ORs in suppl Tables where relevant

We have regenerated these plots so that they show ORs on a log scale, as in the other forest plots. Note that Fig S1 has now been moved to the main text. All tables now show ORs in addition to, or instead of, log ORs.

*index of suppl tables say Table 13 and 14 include information about 8 types of autoimmune diseases. First, many of the diseases are not considered autoimmune (eg asthma, eczema). Second, the tables does not show 8 diseases, it shows 6 diseases (as IBD, I believe) IBD is pool of UC and CD, and T2D is not shown. Please correct

We have now referred to these outcomes as ‘immune-associated’ diseases given that (as the reviewer states) there are certain outcomes here which are not autoimmune diseases. Furthermore, we have now emphasized that UC and CD are encompassed by IBD (page 9):

‘We also obtained estimates using results from a GWAS of T2D, updated since our previous study²⁶, and seven of the most common immune-associated disease endpoints: asthma, atopic dermatitis and eczema, hypothyroidism, rheumatoid arthritis, inflammatory bowel disease and it’s two subtypes (Crohn’s disease and ulcerative colitis).’

T2D results are included in Supplementary Table 15.

*line 313: 10 of 16 analyses in Table S13 significant. This should be 10 of 14, or else T2D results should be added to the table.

Thank you for spotting this typo – we have now corrected it.

* Suppl tables S13 and S14 are a bit confusing but appears to show results for different age at onset of outcomes? Two lines per disease, where the column with the heading outcome contains “age_10” or adult”. I suspect that the age_10 vs adult refers not to the outcome but rather the exposure instrument? The heading of the column specifying diseases is headed by “exposure” but this should probably be “outcome”. Also: the univariate MR results in Table S13 seems also to be contained in Table S14 - why have a separate S13? (except the column with heading FDR , which appear to be the q-value of the univariate MR (?) – this could also be included in S14 and S13 removed.

We have now updated the Supplementary Table legends to clarify that these columns refer to the exposure instrument. We have also taken the reviewers advice and removed the previous Table S13 and incorporated FDR results into the multivariable table of results.

*In the first paragraph of the discussion, the authors say their results support previous observational studies, but no references are cited. Consider citing at least a few well designed studies supporting this claim.

We have now cited Magnus et al. "Infant Growth and Risk of Childhood-Onset Type 1 Diabetes in Children From 2 Scandinavian Birth Cohorts", "Lamb et al, Dietary Glycemic Index, Development of Islet Autoimmunity, and Subsequent Progression to Type 1 Diabetes in Young Children", and Ferrara-Cook et a. "Excess BMI Accelerates Islet Autoimmunity in Older Children and Adolescents" to address this comment.

*the authors used the terms (child or adult) adiposity and body size in the manuscript, but it is not always clear whether it refers to the same concept or not.

We have now clarified this by referring to our MR estimates as childhood/adult body size throughout the manuscript.

*Please provide reference to BOLT-LMM software, as this is not likely generally known.

A reference for the BOLT-LMM software has now been added to page 7.

Reviewer #3 (Remarks to the Author):

This Mendelian analysis seems to confirm the predictive association of childhood obesity with T1D and suggests it is causative in GWAS cohorts, despite the crude measures of obesity and height. These results are consistent with papers in the literature from a variety of T1D prospectively followed cohorts which are not referenced here. A strength is the inclusion of other autoimmune diseases for comparison. However a major weakness is not using the age at T1D diagnosis as the outcome and I think this publication should await the availability of those data. As it stands it appears that adult weight could be included in the prediction of childhood diabetes and lack of association of adult obesity would be expected.

Thank you for these suggestions. We have firstly added information on age at diagnosis for T1D to page 9 as suggested by other reviewers:

'In terms of age-at-diagnosis, 7,453 T1D meta-analysis cases were diagnosed before 10 years of age, 4,368 between 10 and 20 years old, 3,352 over 20 years old and 400 with missing data) (see Supplementary Table 1 for a breakdown by cohort). As such nearly half of cases included in the meta-analysis had age-at-diagnosis later than 10, the age at which our childhood body size instrument is based on.'

However, analyses stratified by age-at-diagnosis of T1D would be extremely difficult to undertake for this study. It would involve a substantial amount of time and work to undertake a T1D genome-wide association study stratified by age at diagnosis data and involve coordinating an effort with many international research groups. Even though this may ultimately allow us to demonstrate that adult adiposity is a risk factor for late onset T1D, it will not diminish the evidence identified in our

current study where the aim was to investigate the causal influence of childhood adiposity of T1D risk.

Other points

1 It is frustrating to have to read other references in order to get a basic understanding of the cohorts examined

2 in order to assess the results, please provide the prevalence of T1D in the various cohorts and overall prevalence

To address both of these suggestions, we have now included a passage in the text describing these cohorts, plus an additional Supplementary Table 1 with further information on age at diagnosis by cohort (page 8):

'Analyses were then repeated separately in each contributing cohort from this meta-analysis: Illumina genotyped UK samples (3,983 cases and 3,994 controls), Affymetrix genotyped UK samples (1,926 cases and 3,342 controls), Sardinians (1,558 cases and 2,882 controls), Finnish FinnGen samples (4,933 cases and 148,190 controls) and the T1DGC European-ancestry family sample (3,173 affected-offspring trios, analysed by the transmission disequilibrium test).'

3. Please review the current literature on the topic and include those references including some on the relationship of BMI with IA progression and correct the statement that there are no studies

4 There are a number of statements in the manuscript that need references and also some need to be checked to be sure they are appropriate

We have referenced an additional MR paper (Censin et al., Causal relationships between obesity and the leading causes of death in women and men" (2019)) in the introduction which reports analysis of obesity and T1D risk, as well as Magnus et al. "Infant Growth and Risk of Childhood-Onset Type 1 Diabetes in Children From 2 Scandinavian Birth Cohorts" as an observational study. We have now also cited "Lamb et al, Dietary Glycemic Index, Development of Islet Autoimmunity, and Subsequent Progression to Type 1 Diabetes in Young Children", and Ferrara-Cook et al. "Excess BMI Accelerates Islet Autoimmunity in Older Children and Adolescents"

Additionally, we have rephrased a sentence in the introduction as the following:

'One hypothesis, supported by some observational studies^{5,6}, is that the rising prevalence of childhood obesity in an increasingly obesogenic environment⁷⁻⁹, including poor diets with high fat, salt and carbohydrate, may contribute towards early life β -cell fragility and increased susceptibility to T1D¹⁰.

5 it appears that reference to figs 1 a,b and c may refer to ref 11. This is confusing

We have now rewritten this section to avoid confusion (page 3):

'We showed previously that childhood adiposity increases T2D risk when analysed in a univariable setting (Odds Ratio (OR)=2.32, 95% confidence interval (CI)=1.76 to 3.05, $P=3.83 \times 10^{-9}$)¹³. This approach to estimate the total effect of childhood body size on risk of disease is presented in Figure 1A. However, by simultaneously estimating the genetically predicted effects of childhood adiposity and adulthood adiposity as separate exposures onto T2D risk using a multivariable model, the childhood estimates attenuated to include the null (OR=1.16, 95% CI=0.74 to 1.82, $P=0.52$). As such, there is considerably weaker evidence that childhood adiposity has a 'direct effect' on T2D risk, as compared to it having an 'indirect effect' mediated via adult adiposity. Diagrams illustrating how multivariable MR can be applied to estimate direct and indirect effect can be found in Figure 1B and Figure 1C respectively. These results therefore suggest that the univariable estimates for childhood adiposity can be explained by long term, persistent effects of adiposity due to individuals typically remaining overweight into adulthood.'

6 There is some literature suggesting that birthweight is related to HLA. Are you suggesting you did not find this?.

In this study our aim was not to investigate whether birthweight is related to HLA. We have added a section to further emphasise that to appropriately investigate the effect of birthweight on disease risk using MR requires a family design which was not possible in this work (page 21):

'However, a more appropriate evaluation of the influence of birthweight on T1D risk requires in-depth evaluation using both maternal and fetal genetic effects, as undertaken previously, once sample sizes of both maternal and offspring T1D cases are sufficient^{36,49}. Amongst other sources of bias, future endeavours applying this study design will be able to investigate whether our results may be underestimating the genetically predicted effect of birthweight on T1D risk.'

Reviewers' Comments:

Reviewer #1:

Remarks to the Author:

I would like to thank the authors for revising their manuscript following my comments. The revisions have addressed my concerns and I have no further comments

Reviewer #2:

Remarks to the Author:

Most of my comments have been addressed, and the revised ms is clearly improved.

A minor comment on calculations of potential impact of change in BMI on population occurrence of T1D: I think we have to assume that the RR will remain constant. A plausible scenario would be to move some (approx 10 of the 16%) who were plumper than average to the average category and leave the thinner than average proportion unchanged. My quick and dirty calculation suggested that this would result in a change in population proportion of T1D from initial 0.5% to approximately 0.43%, a 14% decrease. The calculations in suppl table 14 seems reasonable but I think the assumption there was that the 10% plumper than average who moved category all moved to the thinner than average category? (I may be wrong, but if not this should at least be clarified, or changed).

Reviewer #3:

Remarks to the Author:

The authors have responded to the reviewers comments in the revised manuscript

REVIEWERS' COMMENTS

Reviewer #1 (Remarks to the Author):

I would like to thank the authors for revising their manuscript following my comments. The revisions have addressed my concerns and I have no further comments

Reviewer #2 (Remarks to the Author):

Most of my comments have been addressed, and the revised ms is clearly improved.

A minor comment on calculations of potential impact of change in BMI on population occurrence of T1D: I think we have to assume that the RR will remain constant. A plausible scenario would be to move some (approx 10 of the 16%) who were plumper than average to the average category and leave the thinner than average proportion unchanged. My quick and dirty calculation suggested that this would result in a change in population proportion of T1D from initial 0.5% to approximately 0.43%, a 14% decrease. The calculations in suppl table 14 seems reasonable but I think the assumption there was that the 10% plumper than average who moved category all moved to the thinner than average category? (I may be wrong, but if not this should at least be clarified, or changed).

Many thanks for your further comment to help refine our manuscript. We agree that this point should be clarified and have therefore made the following changes on page 9 of the manuscript:

'Mimicking an intervention and assuming a constant RR, we reduced the proportion of individuals who are 'plumper than average' from 0.159 (i.e. the proportion within our data) to 0.059, and increased the proportion in the 'slimmer than average' category from 0.33 to 0.43. This was to reflect a simplified but realistic scenario in which 10% of individuals move from the high weight to the average weight category, and the same number of average weight individuals move into the low weight category. Our intervention model produced a fall in T1D prevalence to 0.39% (a 22% reduction) (Supplementary Table 15).'

Reviewer #3 (Remarks to the Author):

The authors have responded to the reviewers comments in the revised manuscript